# Geometric Generative Modeling with Noise-Conditioned Graph Networks

**Peter Pao-Huang** [1]   **Mitchell Black** [2]   **Xiaojie Qiu** [1 3]

## Abstract

Generative modeling of graphs with spatial structure is essential across many applications from computer graphics to spatial genomics. Recent flow-based generative models have achieved impressive results by gradually adding and then learning to remove noise from these graphs. Existing models, however, use graph neural network architectures that are independent of the noise level, limiting their expressiveness. To address this issue, we introduce *Noise-Conditioned Graph Networks* (NCGNs), a class of graph neural networks that dynamically modify their architecture according to the noise level during generation. Our theoretical and empirical analysis reveals that as noise increases, (1) graphs require information from increasingly distant neighbors and (2) graphs can be effectively represented at lower resolutions. Based on these insights, we develop Dynamic Message Passing (DMP), a specific instantiation of NCGNs that adapts both the range and resolution of message passing to the noise level. DMP consistently outperforms noise-independent architectures on a variety of domains including 3D point clouds, spatiotemporal transcriptomics, and images. Code is available at https://github.com/peterpaohuang/ncgn.

## 1. Introduction

Geometric graphs, defined by both features and positional information, are a powerful representation in many scientific domains. In computer vision and graphics, these graphs represent 3D point clouds and shapes, enabling generative applications like shape synthesis (Nash et al., 2020) and point cloud generation (Yang et al., 2019). In molecular biology, atoms or amino acids serve as nodes with 3D coordinates while chemical properties constitute the features, accelerating the design of novel drugs (Gómez-Bombarelli et al., 2018; Bengio et al., 2021) and predicting the structure of proteins (Watson et al., 2023; Jumper et al., 2021). In emerging fields like spatial single-cell genomics, geometric graphs capture both the spatial distribution of cells and their gene expression profiles, providing insights into tissue organization and development (Qiu et al., 2024).

To model the distribution of these geometric graphs, recent advances in generative modeling, particularly flow-based approaches such as diffusion (Ho et al., 2020; Song & Ermon, 2019) and flow-matching (Lipman et al., 2022; Tong et al., 2023) models, have shown remarkable success in many scientific domains (Jing et al., 2023; Pao-Huang et al., 2023; Luo et al., 2022). These methods work by learning an iterative transformation between a simple prior distribution (typically Gaussian) and the target data distribution. Their effectiveness can be partially attributed to the surprisingly simple training objective: transform or interpolate the data distribution towards a Gaussian and then learn to reverse this transformation. Colloquially, the interpolation towards a Gaussian can be called a *noising* processing.

While existing graph neural networks for these flow-based generative modeling have demonstrated promising results, they rely on a *fixed* (aka noise-independent) graph neural network architecture across the noising process. For instance, previous works (Corso et al., 2022; Luo et al., 2021; Jing et al., 2022) construct message-passing operations using a constant radius or $k$-nearest neighbor graph independent of the noise level. This static approach limits the ability to fully capture how the underlying graph signal evolves under different noise levels in the generative process.

**Contributions**. This limitation motivates our investigation into the behavior of geometric graphs under varying noise levels. Through an information-theoretic analysis, we prove that as noise increases, recovering the underlying graph signal requires aggregating information from increasingly distant neighbors. This theoretical finding has two important implications. First, message-passing architectures should increase their connectivity radius with the noise level since local structure becomes less informative with increasing noise. We show that this holds in practice by analyzing the

[1]Department of Computer Science, Stanford University [2]Department of Computer Science, University of California San Diego [3]Department of Genetics, Stanford University. Correspondence to: Peter Pao-Huang <peterph@cs.stanford.edu>, Xiaojie Qiu <xiaojie@stanford.edu>.

*Proceedings of the 42$^{nd}$ International Conference on Machine Learning*, Vancouver, Canada. PMLR 267, 2025. Copyright 2025 by the author(s).

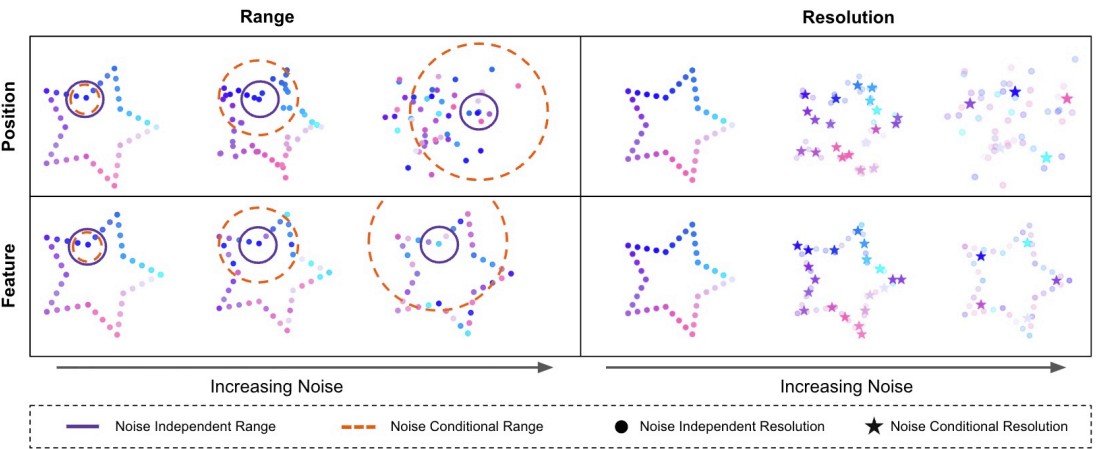

*Figure 1.* Comparing the range and resolution between noise-independent graph neural networks and noise-conditioned graph neural networks (GNNs). (**Top Row**) Noise is introduced in the positions of the star while features are fixed. (**Bottom Row**) Noise is introduced in the features of the star while positions are fixed. (**Left Column**) As noise increases, the range of noise-independent GNNs remains the same while noise-conditional GNNs increases. (**Right Column**) As noise increases, the resolution of noise-independent GNNs remains fixed (shown as circular nodes) while noise-conditioned GNNs decreases (coarse-grained nodes shown as star nodes).

weights of a trained graph attention network used within a flow-based generative model. Second, at higher noise levels, coarse-graining through pooling operations can better preserve the graph signal while reducing computational cost. We also show this empirically by measuring the discrepancy between a dataset of geometric graphs and their noised, coarse-grained counterparts.

Leveraging these theoretical and empirical insights, we propose *Noise-Conditioned Graph Networks* (NCGNs), a new class of graph neural networks that dynamically adapts the graph structure based on the noise level of the generative model. We then develop a specific instantiation of NCGNs called Dynamic Message Passing (DMP). DMP interpolates the range and resolution of the message passing graph from a fully connected low-resolution graph at high noise to a sparsely connected high-resolution graph at low noise. By simultaneously adjusting both connectivity and node resolution according to the noise level, DMP achieves better expressivity than noise-independent architectures while maintaining linear-time message passing complexity.

Through extensive experiments, we demonstrate the effectiveness of our approach across multiple domains. On the ModelNet40 3D shape generation task, DMP achieves a 16.15% average improvement in Wasserstein distance compared to baselines. We further validate our method on a simulated spatiotemporal transcriptomics dataset, where DMP often outperforms existing approaches across multiple biologically inspired generation tasks. Finally, we show that state-of-the-art image generative models can be easily modified to incorporate DMP through minimal code changes, leading to significant improvements in FID with the same computational cost.

## 2. Background

### 2.1. Geometric Generative Modeling

In this work, we focus on modeling geometric graphs, which extend traditional graphs by incorporating spatial information. Formally, we define a geometric graph as a tuple $G = (X, R, \mathcal{E})$ where $X = \{x^{(i)} \in \mathbb{R}^f\}_{i=1}^N$ are node features, $R = \{\eta^{(i)} \in \mathbb{R}^d\}_{i=1}^N$ are spatial positions (in this paper, $d \in \{1, 2, 3\}$), and $\mathcal{E} \subset \binom{[N]}{2}$ are the edges, where $N$ is the number of nodes. In this paper, the edges $\mathcal{E}$ are a function of the positions $R$; for example, they might be the edges of the $k$-nearest neighbor graph. We additionally constrain our definition of geometric generative modeling to learning either the positional distribution conditioned on features $p(R|X)$ or the feature distribution conditioned on positions $p(X|R)$.

This general learning problem has applications in many fields. For instance, quantum systems can be represented by treating particles as nodes with quantum numbers as features and spatial coordinates as positions. Climate science applications represent weather stations as nodes with meteorological measurements as features. Even traditional computer vision tasks can be reformulated under this geometric paradigm by treating pixels as nodes on a regular lattice with RGB features.

### 2.2. Flow-Based Generative Models

A common paradigm to model such distributions is through the unifying lens of flow-based generative models which encompasses diffusion models, flow-matching, and other variants (Liu et al., 2022; Neklyudov et al., 2023; Albergo et al., 2023). Suppose $Z \in \{X, R\}$ and $G_t = (X_t, R, \mathcal{E})$ when

$Z = X$ and $G_t = (X, R_t, \mathcal{E})$ when $Z = R$. These models aim to learn the time-varying vector field $dZ_t = v_\theta(G_t, t)dt$ where $t \in [0, 1]$, which transports a prior distribution $\rho_0(Z)$ (e.g. a Gaussian distribution) to the data distribution $\rho_1(Z)$ such that it satisfies the continuity equation $\frac{\partial p}{\partial t} = -\nabla \cdot (p_t v_\theta(t))$.

We learn $v_\theta(G_t, t)$ with a graph neural network by regressing against the ground truth vector field $u_t(Z_t)$ where $Z_t \sim \mathcal{N}(Z_t | \mu_t(Z_0, Z_1), \sigma_t^2)$ with $Z_0 \sim p_0(Z)$ drawn from the prior and $Z_1 \sim p_1(Z)$ drawn from the data distribution. Intuitively, $u_t$ represents the vector field that guides the noised sample $Z_t$ toward its unnoised target $Z_1$. $\mu_t$ and $\sigma_t^2$ are the mean and variance at noise level $t$. Different flow-based models specify different forms for $\mu_t$, $\sigma_t^2$, and $u_t$; we refer readers to Table 1 of Tong et al. (2023) that details these different forms. In the rest of this work, we will refer to noise level and time $t$ interchangeably (where $t$ close to 1 is low noise and $t$ close to 0 is high noise).

## 2.3. Fixed Graph Structure

To learn $v_\theta(G_t, t)$ with a graph neural network, previous works (Xu et al., 2022; Corso et al., 2022; Luo et al., 2021) construct a fixed (aka *noise-independent*) graph structure from the positions. For instance, the message passing edges in Torsional Diffusion (Jing et al., 2022) are based on a constant-radius graph around node positions. In the case of positional generation, the message passing edges are redrawn for each $t$ based on the noised positions; however, the radius itself or $k$ in $k$-NN are still a constant value across $t$. While one might consider using fully connected architectures like transformers as an alternative (Abramson et al., 2024), these approaches suffer from over-smoothing (Wu et al., 2024) and quadratic computational complexity.

## 3. Gaussianized Geometric Graphs

Through theoretical and empirical analysis, we aim to show that the graph neural network structure should adapt as we change the amount of noise in a geometric graph. To do so, we investigate the behavior of geometric graphs under varying levels of Gaussianization (aka the progressive movement from the data distribution to a Gaussian) either in the graph's positions or features. Specifically, we look at two important components in graph neural networks under noise: the range of connectivity and resolution of nodes.

**Preliminaries**. Under the framework of flow-based generative models, the node positions or features of a graph can be approximated as an isotropic multivariate Gaussian $\mathcal{N}(Z_t, \text{diag}(\sigma_t^2))$. We denote $\sigma_1^2$ as the data distribution's variance. Because the covariance matrix is diagonal, we can treat each node as independent univariate Gaussians $p(z_t^{(i)}) := \mathcal{N}(z_t^{(i)}, \sigma_1^2 + \sigma_t^2)$. The target objective as de-scribed in Section 2 is to recover the unnoised vectors $z_1^{(i)}$ given the noised vectors $z_t^{(i)}$.

## 3.1. Theoretical Analysis

To enable tractable analysis, we introduce several technical assumptions (detailed in Appendix A). However, the results we prove continue to hold empirically even when these assumptions are relaxed, as we demonstrate in Section 3.2. For brevity, we defer all proofs to Appendix A.

### 3.1.1. NOISING OVER FEATURES

To recover the target (i.e. unnoised) features from noised features, we ideally want to maximize the mutual information between the original feature $x_1^{(i)}$ and the accessible information to the graph neural network, which is the aggregation of noisy features. Here, we denote the aggregation of noisy features as $Y_t^{(i,r)}$, which is a function of the radius of aggregation $r$ around node $i$ at noise level $t$.

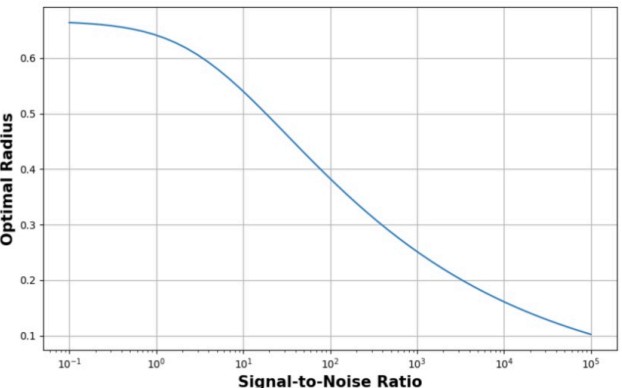

*Figure 2.* **Optimal Radius for Different Noise Levels**. Using the correlation $\rho(\eta^{(i)}, \eta^{(j)}) = 1 - (\eta^{(i)} - \eta^{(j)})^2$, we plot the radius that maximizes mutual information for each signal-to-noise ratio using the formula from Lemma 3.1.

For our first theoretical result, we assume that the positions of the nodes are fixed and we only noise the features.

**Lemma 3.1.** *Let the signal-to-noise ratio be* $\text{SNR}(t) = \frac{\sigma_1^2}{\sigma_t^2}$ *where $\sigma_1^2$ is the initial variance and $\sigma_t^2$ is the variance at $t$, and assume $\text{SNR}(t) > 0$ for all $t$. Let $\rho(\eta^{(i)}, \eta^{(j)})$ be the correlation between the features of node $i$ and $j$ based on their positions. Further, let $D_{r^{(i)}}$ be the closed ball of radius $r$ around node $i$ based on its position $\eta^{(i)}$. The mutual information $I(x_1^{(i)}, Y_t^{(i,r)})$ between clean node feature $x_1^{(i)}$ and the aggregation $Y_t^{(i,r)}$ of noised features is*

$$I(x_1^{(i)}, Y_t^{(i,r)}) = \frac{1}{2} \log \left( \frac{\frac{2r}{\text{SNR}(t)} + A}{\frac{2r}{\text{SNR}(t)} + A - B^2} \right)$$

*where*

$$A = \int_{j,k \in D_{r^{(i)}}} \rho(\eta^{(j)}, \eta^{(k)})djdk \,,$$

$$B = \int_{j \in D_{r^{(i)}}} \rho(\eta^{(i)}, \eta^{(j)})dj \,.$$

This lemma provides a formula for calculating the mutual information between the original signal $x_1^{(i)}$ and the aggregated noisy signal $Y_t^{(i,r)}$ based only on the correlation $\rho$.

If we assume $\rho(\eta^{(i)}, \eta^{(j)}) = 1 - (\eta^{(i)} - \eta^{(j)})^2$, Figure 2 visualizes the mutual information (calculated using Lemma 3.1). Clearly, increasing the signal-to-noise ratio causes the radius with maximum mutual information to decrease, and decreasing the signal-to-noise ratio causes the radius that achieves maximum mutual information to increase. This relationship is formalized with the following theorem:

**Theorem 3.2.** *Assume that $\rho(\eta^{(i)}, \eta^{(j)}) = 1 - (\eta^{(i)} - \eta^{(j)})^2$ where $0 < r \le 1$. Let $r_1$ be the radius that maximizes mutual information for a given signal-to-noise ratio $c_1$. If we decrease the signal-to-noise ratio $c_2 < c_1$, then there exists $r_2 > r_1$ such that*

$$I(x_1^{(i)}, Y_{c_2}^{(r_2)}) > I(x_1^{(i)}, Y_{c_2}^{(r_1)}) \,.$$

Since aggregation is a simplified form of message passing, Theorem 3.2 suggests that message passing architectures should adapt their connectivity pattern with noise level. At high noise, nodes can increase information by communicating with a broader neighborhood, while at low noise, local connectivity is optimal for maximizing information.

Importantly, aggregation can also be viewed as a form of pooling or coarse-graining. Both operations aggregate information over a local neighborhood. Theorem 3.2 therefore also provides theoretical justification for why reducing graph resolution becomes advantageous at higher noise levels. By pooling nodes together, we perform a form of aggregation that helps denoise the graph while preserving essential structural information.

### 3.1.2. NOISING OVER POSITIONS

Another common case is learning to reverse the noise over positions. However, we cannot follow the same proof structure as the previous case, as the aggregation of nodes is now also dependent on the noise level rather than just the radius of aggregation; as data becomes more Gaussian, the number of nodes in some radius of node $i$ is dependent on the positional center of node $i$. Instead, Proposition 3.3 proves that as we increase the noise level, the expected distance between any close pair of nodes increases. Therefore, the radius of message passing must increase, so that originally nearby nodes can communicate.

**Proposition 3.3.** *Let $||\eta_1^{(i)} - \eta_1^{(j)}||_2 = \gamma << 1$. Given variance-preserving noise and correlation structure between positions as $\rho$, the expected squared distance between node $i$ and $j$ with respect to noise level $t$ is*

$$\mathbb{E}[(\eta_t^{(i)} - \eta_t^{(j)})^2] = 2(1-t) + 2\rho(\gamma)(1-t) + t\gamma^2 \,.$$

While this proposition offers a more elementary treatment than the analysis of noising over features, it reinforces the notion that the radius must increase with the noise level.

### 3.2. Empirical Analysis

To validate that the theoretical analysis holds in practice without the simplifying assumptions, we conduct an empirical demonstration on a dataset of 3D geometric objects (Fey & Lenssen, 2019) (spheres, cubes, triangular prisms, etc).

**Increasing Noise → Increasing Connectivity**. We train two separate $v_\theta$ networks on two separate tasks: (1) generate the positions of the geometric objects and (2) generate the features of the geometric objects. Since this dataset does not come with node features, we construct features $x_i$ for each node $i$ with position $\eta_i$ by computing the displacement vector from the node to the shape's center of mass $\mu$: $x_i = \eta_i - \mu$ where $\mu = \frac{1}{N} \sum_{j=1}^N \eta_j$.

The network $v_\theta$ is a fully connected single-layer graph attention network. The goal of this task is to show what portions (and more importantly, what ranges) of the geometric object are relevant to the graph attention network at different levels of noise.

Figure 3 shows the average (over the dataset) attention weight distribution across noise levels. At low noise, both position and feature generation models focus primarily on nearby nodes. As noise increases, both models shift attention to more distant nodes, but in distinctive ways: position generation becomes sharply focused on distant nodes while ignoring nearby ones, whereas feature generation maintains a more uniform attention distribution across all distances.

These empirical attention patterns support Theorem 3.2 and Proposition 3.3. As noise increases, the radius of effective communication must increase. However, this does not necessitate a fully connected graph, which would incur *quadratic complexity* and problems of oversmoothing. Instead, our theory suggests we can achieve optimal performance by dynamically structuring connectivity based on the noise level, connecting only nodes within the relevant radius.

**Increasing Noise → Decreasing Resolution**. To understand how noise affects the optimal graph resolution, we conduct a systematic analysis of coarse-graining effects. For each geometric shape in our dataset, we (1) generate versions with different noise levels, (2) create coarse-grained representations using $K$-means clustering with varying num-

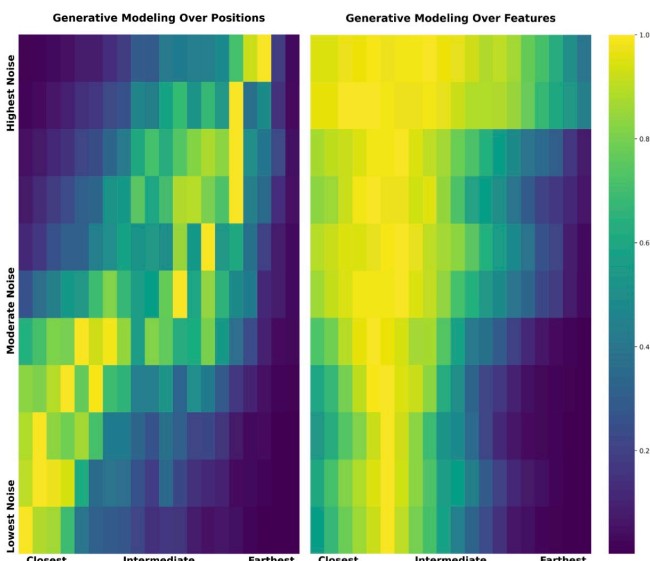

*Figure 3.* **Attention Distribution over Node Distances**. Average attention weight distribution between pairs of nodes as a function of their spatial distance, shown for different noise levels in a graph attention network.

bers of clusters, and then (3) compare each noised and coarse-grained version to its original, unnoised shape.

To quantify the similarity between shapes, we adopt the Gromov-Wasserstein (GW) distance from object registration literature (Mémoli, 2011). Given two point clouds, the GW distance measures how well their geometric structures match while being invariant to rigid transformations (i.e. a lower GW distance signifies a better match).

Figure 4 shows how the optimal level of coarse-graining varies with noise level, where optimality is defined as minimizing the Gromov-Wasserstein distance to the original shape. It is clear that as noise increases, the optimal number of clusters decreases. This pattern indicates that higher levels of coarse-graining (fewer clusters) better preserve the underlying geometric structure when noise is high. This relationship holds across different pooling operations (max and mean) and across noising the node features or positions.

## 4. Noise-Conditioned Graph Networks

Let $v_\theta : G \times [0,1] \to Z$ be the graph neural network used to predict the vector field $dZ = v_\theta(G_t, t)dt$. Previous work uses a fixed network independent of noise level $t$ as described in Section 2.3:

$$v_\theta(G_t, t) = \text{GNN}(G_t) .$$

Here, GNN is an abstract message passing network with flexible design choices like structural elements (edge connectivity) and algorithmic components (message passing functions, aggregation schemes, and node update rules).

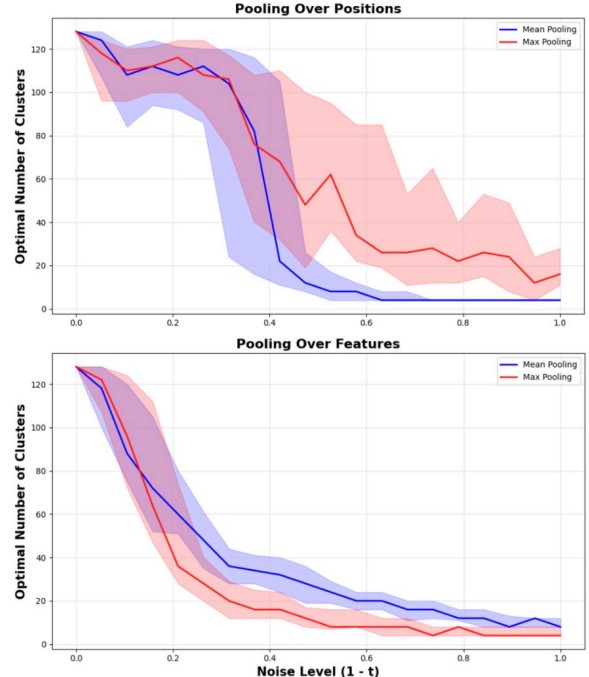

*Figure 4.* **Optimal Graph Resolution at Different Noise Levels**. For each noise level, we plot the number of clusters that minimizes the Gromov-Wasserstein distance between the coarse-grained noisy graph and the original graph. Results are plotted separately for max and mean pooling operations.

Our key insight from Section 3 is that GNN should adapt to the noise level $t$. This leads us to propose Noise-Conditioned Graph Networks (NCGNs):

$$v_\theta(G_t, t) = \text{NCGN}_t(G_t) .$$

The core difference here is that the previously static network GNN is now conditioned on the noise level as $\text{NCGN}_t$. For example, $\text{NCGN}_t$ may use a different architecture for different values of $t$ or may perform message passing over the $k_t$ nearest neighbors, where $k_t$ is a function of $t$. While $\text{NCGN}_t$ is intentionally general, in the following section, we develop a specific instantiation of NCGNs that dynamically changes its message passing connectivity and resolution according to the noise level of the generative process.

### 4.1. Dynamic Range & Resolution Message Passing

We introduce an instance of NCGN called Dynamic Message Passing (DMP). DMP specifically modifies the message passing connectivity (i.e. the edges in the message passing graph) and the message passing resolution (how information is propagated through intermediary nodes). For the latter, DMP introduces coarse-grained "supernodes" that serve only to relay messages between different parts of the graph during the message passing phase. Importantly, the network output $v_\theta$ remains defined over all original nodes;

the coarse-graining only affects how messages are propagated internally.

Let $r_t$ be the range (e.g. kNN connectivity or radius graph) and $s_t$ the number of supernodes used for message passing at noise level $t$. Further, suppose we are given boundary conditions $(r_0, s_0)$ and $(r_1, s_1)$ that specify the connectivity and resolution at $t = 0$ and $t = 1$ respectively. DMP defines NCGN$_t$ solely based on $r_t$ and $s_t$. To determine $r_t$ and $s_t$, we use an *adaptive scheduler*,

$$r_t, s_t = f(t, r_0, r_1, s_0, s_1).$$

Building on Section 3, we require the scheduler $f$ to satisfy the following constraints:

(**Monotonicity**) For $t' < t$ (where $t'$ represents higher noise), we must have $r' \geq r$ and $s' \leq s$ where $r', s' = f(t', r_0, r_1, s_0, s_1)$. This follows from Section 3, which shows that increasing noise requires a greater radius for information aggregation and reduced graph resolution through coarser-grained nodes.

(**Boundary Condition Satisfaction**) $r_1, s_1 = f(1, r_0, r_1, s_0, s_1)$ and $r_0, s_0 = f(0, r_0, r_1, s_0, s_1)$.

Since deriving an optimal scheduler $f$ is unclear, we rely on a predefined $f$ that satisfies the above conditions. Some examples of $f$ are shown in Figure 5. Section 5 provides guidance on which schedule performs best in practice.

**Graph Construction**. With a specific $r_t$ and $s_t$ given by $f$, DMP implements NCGN$_t$ with the following steps:

Graph Coarsening. DMP first maps the noisy graph $G_t$ to a coarse representation based on $s_t$ as

$$\hat{X}, \hat{R}, \hat{\mathcal{E}} = \mathrm{coarsen}(s_t, X, R, \mathcal{E})$$

where $|\hat{R}| = |\hat{X}| = s_t$. For instance, coarsen can be a deterministic mean pooling or a learnable message passing layer from original to coarse-grained nodes. This coarse representation serves exclusively as an intermediate structure for efficient message passing, while $v_\theta$ maintains its output mapping over the complete set of original nodes.

Edge Structure. Given coarse-grained edges $\hat{\mathcal{E}}$, DMP defines additional message passing edges between coarse-grained nodes according to the connectivity range $r_t$:

$$\hat{\mathcal{E}} = \mathrm{concat}(\hat{\mathcal{E}}, \mathrm{constructEdges}(\hat{R}, r_t))$$

where constructEdges implements a proximity-based connectivity pattern such as a kNN or radius graph.

Message Passing Step. Rather than propagating messages between original nodes, DMP performs message passing over the coarse-grained representation:

$$\hat{X}, \hat{R} = \mathrm{MP}(\hat{X}, \hat{R}, \hat{\mathcal{E}}).$$

---

**Algorithm 1** Forward Pass of DMP

1: **Input:** $G_t$, $t$, $r_0$, $r_1$, $s_0$, $s_1$
2: **Output:** $v_\theta^{(\mathrm{pred})}$
3: $r_t, s_t \leftarrow f(t, r_0, r_1, s_0, s_1)$
4: $X, R \leftarrow \mathrm{lift}(X, R)$
5: **for** $k = 1$ **to** $K$ GNN layers **do**
6: $\quad \hat{X}, \hat{R}, \hat{\mathcal{E}} \leftarrow \mathrm{coarsen}_k(s, X, R, \mathcal{E})$
7: $\quad \hat{\mathcal{E}} \leftarrow \mathrm{concat}(\hat{\mathcal{E}}, \mathrm{constructEdges}(\hat{R}, r))$
8: $\quad \hat{X}, \hat{R} \leftarrow \mathrm{MP}_k(\hat{X}, \hat{R}, \hat{\mathcal{E}})$
9: $\quad X, R \leftarrow \mathrm{uncoarsen}_k(\hat{X}, \hat{R}, X, R, \hat{\mathcal{E}}, \mathcal{E})$
10: **end for**
11: $v_\theta^{(\mathrm{pred})} \leftarrow \mathrm{proj}(X, R)$

---

Uncoarsening Step. Finally, since $v_\theta$ operates on the full graph resolution, DMP maps the processed information back to the original nodes:

$$X, R = \mathrm{uncoarsen}(\hat{X}, \hat{R}, X, R, \hat{\mathcal{E}}, \mathcal{E}).$$

This operation can be the original nodes aggregating from their associated coarse-grained nodes. uncoarsen could also be another learnable message passing layer where the coarse-grained node message passes to the original nodes.

**Minimal Implementation Effort**. Converting an existing GNN architecture to DMP requires minimal modification. The primary changes involve implementing the coarsening and uncoarsening operations, while the core GNN architecture remains unchanged, operating on the coarse-grained representation. This modification also preserves all other aspects of the flow-based generative model. We show this in practice by modifying an existing state-of-the-art generative model to DMP in Section 5.3.

**Linear Time Complexity.** DMP can achieve linear time complexity while maintaining expressiveness through the choice of boundary conditions and the constructEdges operation. Consider a setup where at no noise ($t = 1$) we have full resolution ($s_1 = N$ nodes) with sparse connectivity ($r_1$ neighbors per node), while at high noise ($t = 0$) we have a coarse resolution ($s_0 = \sqrt{r_1 N}$ nodes) with full connectivity. When $f$ interpolates between these boundaries such that $r_t s_t = r_1 N$ (where $r_t$ is the number of neighbors and $s_t$ is the number of coarse-grained nodes at time $t$), message passing maintains linear time complexity throughout the generation process.

## 5. Experiments

We evaluate the DMP instantiation of NCGNs across three settings: generating geometric graphs over *positions*, generating geometric graphs over *features*, and adapting existing generative models.

**Setup**. For the first two experiments, we implement DMP

| MODELNET40 | RANDOM | KNN | LONG-SHORT RANGE | DMP |
|---|---|---|---|---|
| GCN | 8.624 | 5.882 | 4.315 | **4.215** |
| GAT | 8.624 | 5.598 | 4.741 | **4.263** |

*Table 1.* Wasserstein distance ($\times 10^{-2}$) between the predicted distribution and the ground distribution of the ModelNet40 test set (the lower the better).

| MODELNET40 | $\mathcal{W}_2$ ($\times 10^{-2}$) |
|---|---|
| LINEAR | 4.495 |
| EXPONENTIAL | **4.263** |
| LOGARITHM | 4.860 |
| RELU | 4.485 |

*Table 2.* Comparing different adaptive schedules for DMP. Each result is reported on flow-matching models with GAT as $v_\theta$.

in the same way. We have 3 message passing layers where we evaluate two popular message passing architectures, convolutions (GCNs) (Kipf & Welling, 2016) and attention (GATs) (Veličković et al., 2017), to show that in either case, our method can still perform well. The coarsening operation is a mean over the nodes in the cluster where the clusters are determined by voxel clustering. Our boundary conditions and edge construction match that of "Linear Time Complexity" in Section 4.1. As such, the DMP implementations in these two experiments have *linear time complexity*.

We use DMP within two classes of flow-based generation models: diffusion models (Ho et al., 2020) in the first experiment and flow-matching models (Tong et al., 2023) in the second experiment. For diffusion models, we use 1000 number of function evaluations (NFEs), and for flow-matching models, we use 200 NFEs. All samples (both positions and features) are normalized to $[-0.5, 0.5]$. The rest of our hyperparameters are recorded in Appendix B.2 and fixed between both experiments; further experiment details including the DMP architecture can be found in Appendix B.

For the third experiment, we demonstrate how an existing flow-based generative model can be easily adapted to incorporate noise-conditioned graph networks by modifying a state-of-the-art image generation model: Diffusion Transformer (DiT (Peebles & Xie, 2023)). We denote the resulting modified architecture as DiT-DMP.

### 5.1. Generating 3D Objects

**Dataset**. DMP is evaluated on ModelNet40 (Wu et al., 2015), a dataset of 3D CAD models for everyday objects (e.g. airplanes, beds, benches, etc) with 40 classes in total. From each object, we sample 400 points and treat the points as a point cloud. The training and testing set consists of 9843 and 2232 objects, respectively. On this dataset, we aim to learn the distribution over *positions* of the point clouds.

**Baselines**. We evaluate DMP against two baselines: (1) a full resolution (i.e. no coarse-graining) point cloud with message passing edges constructed based on a $k$NN neighborhood and (2) a full resolution point cloud with message passing edges constructed with both short and long-range edges (denoted as "long-short range"). For each node in the long-short range baseline, we uniformly sample $k$ edges.

It is particularly important to test against long-short range because we aim to show that the performance benefit of DMP does not come from having both long and short range message passing, but rather that at each noise level, it is beneficial to only look at a specific neighborhood range. Additionally, DMP uses the exponential function as the adaptive schedule for this experiment. However, we also investigate the performance of other adaptive schedules in Section 5.1.1

**Results**. We evaluate DMP and the baselines using the Wasserstein ($\mathcal{W}_2$) distance between the test set distribution and the generated distribution. The results are shown in Table 1. Across both GCN and GAT message passing architectures, we see a consistent decrease of $16.15\%$ in $\mathcal{W}_2$ distance on average by simply having a noise-conditioned graph network rather than the fixed form of baselines.

#### 5.1.1. COMPARING ADAPTIVE SCHEDULES

From Section 4.1, it is clear that many functional forms satisfy the constraints for the adaptive schedule. To investigate the impact of different adaptive schedules, we compare the performance between the schedules shown in Figure 5 on ModelNet40 using GAT as the message passing architecture and diffusion as the generation model. Table 2 displays the results for each schedule.

As shown in the table, while there is variability in performance between schedules, all schedules except the logarithm schedule still perform better than the baselines. Based on this experiment, since the exponential schedule performed best, we will continue using only the exponential schedule for all following experiments.

### 5.2. Generating Spatiotemporal Transcriptomics

**Dataset**. We evaluate on a spatiotemporal transcriptomics dataset simulating gene expression patterns across both space and time in developing limbs. This is an important task with applications in fundamental biological discovery (Tong et al., 2020; Qiu et al., 2022) and therapeutics (Bunne et al., 2023). Each node in the graph represents a cell with features corresponding to the expression levels of three genes (Bmp, Sox9, and Wnt), while node positions encode both temporal progression and spatial location in the

| $\mathcal{W}_2$ | RANDOM | KNN | FULLY CONNECTED | DMP |
|---|---|---|---|---|
| UNCONDITIONAL GENERATION | 2.452 | 2.030/0.840 | 0.880/1.094 | **0.812/0.834** |
| TEMPORAL TRAJECTORY PREDICTION | 2.681 | 2.290/**1.022** | 1.060/1.271 | **0.960**/1.032 |
| TEMPORAL INTERPOLATION | 3.982 | 3.390/2.946 | 2.923/3.026 | **2.880/2.912** |
| GENE IMPUTATION | 2.810 | 2.943/**2.527** | **2.507**/2.534 | 2.529/2.575 |
| SPACE IMPUTATION | 3.344 | 3.222/2.928 | 2.923/2.946 | **2.907/2.913** |
| GENE KNOCKOUT | 3.202 | 2.890/**2.390** | 2.424/2.473 | **2.392**/2.397 |

*Table 3.* Wasserstein distance between the predicted distribution and the ground distribution of the spatial transcriptomic dataset (GCN performance / GAT performance).

| IMAGENET $256 \times 256$ | DiT | DiT-DMP |
|---|---|---|
| FID↓ | 84.051 | **63.983** |
| sFID↓ | 18.787 | **12.823** |
| IS↑ | 16.735 | **24.681** |
| PRECISION↑ | 0.296 | **0.446** |
| RECALL↑ | 0.394 | **0.426** |

*Table 4.* Performance between DiT and its modified DMP version.

tissue. Following Raspopovic et al. (2014), we simulate the data using a reaction-diffusion system that captures the key gene regulatory dynamics in limb formation. The training set contains 10000 samples with 100 nodes each, while the test set uses a different discretization with 96 nodes to evaluate model robustness to varying resolutions. Detailed data generation procedures and biological context are provided in Appendix B.3.

**Baselines**. We compare DMP to the following baselines: (1) a kNN neighborhood graph and (2) a fully connected graph on important biologically inspired tasks. Note that unlike Section 5.1, we only have 100 nodes for each graph, so it is computationally feasible to baseline against a fully connected graph.

Because we found from the previous experiment that the exponential adaptive schedule works best, we proceed with only using the exponential schedule. To showcase that DMP is compatible with other flow-based generative models outside of diffusion models, we will use the more recent formulation of flow-matching in this experiment.

**Results**. Summarized in Table 3, we see that DMP has lower $\mathcal{W}_2$ distance on GCN and comparable performance on GAT against baselines. Especially for specific tasks like gene imputation or gene knockout, there is not a significant difference in performance between our method and baselines.

### 5.3. Adopting an Existing Generative Model

For a given noised image in the generative process, DiT first encodes the image to a $32 \times 32$ latent representation using an autoencoder. Then, DiT partitions this representation into patches of size $p \in \{2, 4, 8\}$, yielding coarse-grained

representations of dimensions $16 \times 16$, $8 \times 8$, or $4 \times 4$ respectively. A transformer then learns to reverse the noise process. We modify DiT's architecture to implement DMP through two simple changes: (1) we adopt FlexiViT's (Beyer et al., 2023) dynamic patch sizing to increase patch sizes at higher noise levels, and (2) we replace the transformer's quadratic attention mechanism with neighborhood attention (Hassani et al., 2023) where neighborhood size scales with noise level. Detailed implementation specifications are provided in Appendix C.

**Dataset**. Evaluation is conducted on the ImageNet $256 \times 256$ dataset (Deng et al., 2009), composed of 1281167 training images and 100000 test images across 1000 object classes.

**Baselines**. We evaluate baseline DiT using patch size $p = 4$ and the Small model configuration (Peebles & Xie, 2023) with 12 layers, hidden size of 384, and 6 attention heads. Our modified DiT-DMP has the same computational complexity as the baseline.

**Results**. As shown in Table 4, the modified DiT-DMP significantly outperforms the state-of-the-art DiT model across various metrics on unconditional generation. Notably, these gains are achieved while maintaining identical computational complexity to the baseline and requiring only two straightforward code changes: replacing the fixed patch size with FlexiViT's dynamic patch sizing and swapping the transformer's full attention for neighborhood attention.

## 6. Conclusion

We introduced Noise-Conditioned Graph Networks, demonstrating that adapting graph neural network architectures to noise levels can improve performance for geometric generative modeling. We showed that optimal information aggregation requires increasing radius with noise level, while graph resolution can be reduced without losing much information. However, DMP is currently limited to a predefined range and resolution schedule. Future works could introduce learnable range and resolution parameters. Further, several other components in graph neural networks could be adapted outside of range and resolution including the number of layers, message passing type, or width of the

layers.

## Impact Statement

This paper presents work whose goal is to advance the field of Machine Learning. There are many potential societal consequences of our work, none which we feel must be specifically highlighted here

## Acknowledgements

We thank Francesco Di Giovanni, Wenzhuo Tang, Yifan Lu, and anonymous ICML reviewers for helpful feedback and discussions. PPH is supported by the National Science Foundation GRFP and acknowledges the support by the National Institutes of Health DP2 New Innovator Award (1DP2HG014282-01) and MorPhiC consortia grant (5U01HG013176-02). MB is supported by NSF grants CCF-2217058 and CCF-2310411.

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

# A. Proofs

## A.1. Noising over Features

**Assumptions**. We make the following assumptions throughout the analysis of noising over features:

(i) The noising process is defined as $z_t = z_1 + \epsilon_t$ where $\epsilon_t \sim \mathcal{N}(0, \sigma_t^2)$.

(ii) Node positions and features are supported on the real line: $x^{(i)}, \eta^{(i)} \in \mathbb{R}$.

(iii) We treat the graph in a continuous manner rather than a discrete object, meaning that we have a corresponding node $(x^{(i)}, \eta^{(i)})$ for all $i \in \mathbb{R}$, with the position $\eta^{(i)} = i$.

(iv) The message passing scheme is a summation over all node features in some closed ball $D_{r^{(i)}}$ of radius $r$ around node $i$ based on its position $\eta^{(i)}$. This aggregated sum of features $Y_t^{(i,r)} = \int_{j \in D_{r^{(i)}}} x_t^{(j)} dj = \int_{j \in D_{r^{(i)}}} x_1^{(j)} + \epsilon_t^{(j)} dj$ represents the information accessible to node $i$ at noise level $t$.

**Lemma 3.1.** *Let the signal-to-noise ratio be* $\mathrm{SNR}(t) = \frac{\sigma_1^2}{\sigma_t^2}$ *where $\sigma_1^2$ is the initial variance and $\sigma_t^2$ is the variance at $t$, and assume* $\mathrm{SNR}(t) > 0$ *for all $t$. Let $\rho(\eta^{(i)}, \eta^{(j)})$ be the correlation between the features of node $i$ and $j$ based on their positions. Further, let $D_{r^{(i)}}$ be the closed ball of radius $r$ around node $i$ based on its position $\eta^{(i)}$. The mutual information $I(x_1^{(i)}, Y_t^{(i,r)})$ between clean node feature $x_1^{(i)}$ and the aggregation $Y_t^{(i,r)}$ of noised features is*

$$I(x_1^{(i)}, Y_t^{(i,r)}) = \frac{1}{2} \log \left( \frac{\frac{2r}{\mathrm{SNR}(t)} + A}{\frac{2r}{\mathrm{SNR}(t)} + A - B^2} \right)$$

*where*

$$A = \int_{j,k \in D_{r^{(i)}}} \rho(\eta^{(j)}, \eta^{(k)}) djdk \,,$$

$$B = \int_{j \in D_{r^{(i)}}} \rho(\eta^{(i)}, \eta^{(j)}) dj \,.$$

*Proof.* We aim to calculate

$$I(x_1^{(i)}, Y_t^{(i,r)}) = H(x_1^{(i)}) + H(Y_t^{(i,r)}) - H(x_1^{(i)}, Y_t^{(i,r)}).$$

The entropy of Gaussians are known in closed form as

$$H(x_1^{(i)}) = \frac{1}{2} \log \left( 2\pi e \sigma_{x_1^{(i)}}^2 \right)$$

$$H(Y_t^{(i,r)}) = \frac{1}{2} \log \left( 2\pi e \sigma_{Y_t^{(i,r)}}^2 \right)$$

where

$$\sigma_{Y_t^{(i,r)}}^2 = \mathrm{var}(Y_t^{(i,r)}) = \mathrm{var} \left( \int_{j \in D_{r^{(i)}}} x_t^{(j)} dj \right) = \int_{j,k \in D_{r^{(i)}}} \mathrm{cov}(x_t^{(j)}, x_t^{(k)}) djdk \,.$$

Following variance-exploding noise,

$$= \int_{j,k \in D_{r^{(i)}}} \mathrm{cov}(x_1^{(j)} + \epsilon_t^{(j)}, x_1^{(k)} + \epsilon_t^{(k)}) djdk \quad \text{where} \quad \epsilon_t = \mathcal{N}(0, \sigma_t^2)$$

$$= \int_{j \neq k \in D_{r^{(i)}}} \sigma_1^2 \rho(\eta^{(j)}, \eta^{(k)}) djdk + \int_{j=k \in D_{r^{(i)}}} \sigma_1^2 + \sigma_t^2 djdk$$

$$= 2r\sigma_t^2 + \sigma_1^2 \int_{j,k \in D_{r^{(i)}}} \rho(\eta^{(j)}, \eta^{(k)}) djdk$$

and

$$\sigma^2_{x_1^{(i)}} = \sigma^2_1 \,.$$

Further, we have

$$H(x_1^{(i)}, Y_t^{(i,r)}) = \frac{1}{2} \log \left( (2\pi e)^2 (\sigma^2_{x_1^{(i)}} \sigma^2_{Y_t^{(i,r)}} - \sigma^4_{x_1^{(i)}, Y_t^{(i,r)}}) \right)$$

with

$$\sigma^2_{x_1^{(i)}, Y_t^{(i,r)}} = \mathrm{cov}\left( x_1^{(i)}, \int_{j \in D_{r(i)}} x_1^{(j)} + \epsilon_t^{(j)} \mathrm{d}j \right) = \int_{j \in D_{r(i)}} \mathrm{cov}(x_1^{(i)}, x_1^{(j)}) \mathrm{d}j = \sigma^2_1 \int_{j \in D_{r(i)}} \rho(\eta^{(i)}, \eta^{(j)}) \mathrm{d}j \,.$$

Consequently,

$$
\begin{aligned}
I(x_1^{(i)}, Y_t^{(i,r)}) &= \frac{1}{2} \log \left( \frac{\sigma^2_{x_1^{(i)}} \sigma^2_{Y_t^{(i,r)}}}{\sigma^2_{x_1^{(i)}} \sigma^2_{Y_t^{(i,r)}} - (\sigma^2_{x_1^{(i)}, Y_t^{(i,r)}})^2} \right) \\
&= \frac{1}{2} \log \left( \frac{\sigma^2_1 (2r\sigma^2_t + \sigma^2_1 \int_{j,k \in D_{r(i)}} \rho(\eta^{(j)}, \eta^{(k)}) \mathrm{d}j\mathrm{d}k)}{\sigma^2_1 (2r\sigma^2_t + \sigma^2_1 \int_{j,k \in D_{r(i)}} \rho(\eta^{(j)}, \eta^{(k)}) \mathrm{d}j\mathrm{d}k) - (\sigma^2_1 \int_{j \in D_{r(i)}} \rho(\eta^{(i)}, \eta^{(j)}) \mathrm{d}j)^2} \right) \\
&= \frac{1}{2} \log \left( \frac{2r\sigma^2_t + \sigma^2_1 \int_{j,k \in D_{r(i)}} \rho(\eta^{(j)}, \eta^{(k)}) \mathrm{d}j\mathrm{d}k}{2r\sigma^2_t + \sigma^2_1 \int_{j,k \in D_{r(i)}} \rho(\eta^{(j)}, \eta^{(k)}) \mathrm{d}j\mathrm{d}k - \sigma^2_1 (\int_{j \in D_{r(i)}} \rho(\eta^{(i)}, \eta^{(j)}) \mathrm{d}j)^2} \right) \,.
\end{aligned}
$$

Factoring out $\sigma^2_1$,

$$= \frac{1}{2} \log \left( \frac{2r\frac{\sigma^2_t}{\sigma^2_1} + \int_{j,k \in D_{r(i)}} \rho(\eta^{(j)}, \eta^{(k)}) \mathrm{d}j\mathrm{d}k}{2r\frac{\sigma^2_t}{\sigma^2_1} + \int_{j,k \in D_{r(i)}} \rho(\eta^{(j)}, \eta^{(k)}) \mathrm{d}j\mathrm{d}k - (\int_{j \in D_{r(i)}} \rho(\eta^{(i)}, \eta^{(j)}) \mathrm{d}j)^2} \right) \,.$$

Define the signal-to-noise ratio as $\mathrm{SNR}(t) = \frac{\sigma^2_1}{\sigma^2_t}$. Then,

$$I(x_1^{(i)}, Y_t^{(i,r)}) = \frac{1}{2} \log \left( \frac{\frac{2r}{\mathrm{SNR}(t)} + \int_{j,k \in D_{r(i)}} \rho(\eta^{(j)}, \eta^{(k)}) \mathrm{d}j\mathrm{d}k}{\frac{2r}{\mathrm{SNR}(t)} + \int_{j,k \in D_{r(i)}} \rho(\eta^{(j)}, \eta^{(k)}) \mathrm{d}j\mathrm{d}k - (\int_{j \in D_{r(i)}} \rho(\eta^{(i)}, \eta^{(j)}) \mathrm{d}j)^2} \right) \,. \qquad \square$$

**Lemma A.1.** *Assume that the correlation $\rho$ follows the following form where $0 < \gamma < r \le 1$,*

$$\rho(\eta^{(i)}, \eta^{(j)}) = 1 - (\eta^{(i)} - \eta^{(j)})^2.$$

*Then,*

$$I(x_1^{(i)}, Y_t^{(i,r)}) = \frac{1}{2} \log(\kappa(r, c)),$$

*where $\kappa(r, c) = \frac{\frac{2}{c} + 4r - \frac{8r^3}{3}}{\frac{2}{c} + \frac{4r^6}{9}}$ and $c = \mathrm{SNR}(t)$.*

*Proof.* First, we solve for

$$\mathrm{A} = \int_{j,k \in D_{r(i)}} \rho(\eta^{(j)}, \eta^{(k)}) \mathrm{d}j\mathrm{d}k = \int_{j,k \in D_{r(i)}} 1 - (\eta^{(j)} - \eta^{(k)})^2 \mathrm{d}j\mathrm{d}k = 4r^2 - \frac{8r^4}{3}$$

and

$$\mathrm{B} = \int_{j \in D_{r(i)}} \rho(\eta^{(i)}, \eta^{(j)}) \mathrm{d}j = \int_{j \in D_{r(i)}} 1 - (\eta^{(j)})^2 \mathrm{d}j = 2r - \frac{2r^3}{3} \,.$$

Based on Lemma 3.1, we have

$$
\begin{aligned}
I(x_1^{(i)}, Y_t^{(i,r)}) &= \frac{1}{2} \log \left( \frac{\frac{2r}{\text{SNR}(t)} + A}{\frac{2r}{\text{SNR}(t)} + A - B^2} \right) \\
&= \frac{1}{2} \log \left( \frac{\frac{2r}{c} + 4r^2 - \frac{8r^4}{3}}{\frac{2r}{c} + \frac{4r^6}{9}} \right) \\
&= \frac{1}{2} \log \left( \frac{\frac{2}{c} + 4r - \frac{8r^3}{3}}{\frac{2}{c} + \frac{4r^6}{9}} \right)
\end{aligned}
$$

where $c = \text{SNR}(t)$. $\qquad \square$

**Lemma A.2.** *Let $\kappa(r, c)$ and $\rho$ be defined as in Lemma A.1. Assume we restrict the support of $r$ to be $\in [\epsilon, 1]$ where $\epsilon$ is sufficiently small. For a given signal-to-noise ratio $c$ at time $t$ (denoted as $c_t$), there exists a $r^*$ such that*

$$
r^* = \max_{\epsilon \leq r \leq 1} \kappa(r, c) \quad and \quad \frac{\mathrm{d}\kappa(r^*, c_t)}{\mathrm{d}r^*} = 0.
$$

*Proof.* By the extreme value theorem, since the domain is compact ($r \in [\epsilon, 1]$) and $I$ is continuous, $I$ must achieve a maximum within this interval. Further, we need to show that the maximum is achieved at an interior point $r^* \in (\epsilon, 1)$ to prove that $\frac{\mathrm{d}\kappa(r^*, c_t)}{\mathrm{d}r^*} = 0$.

Taking the derivative of $\kappa$ from Lemma A.1,

$$
\frac{\mathrm{d}\kappa}{\mathrm{d}r} = \frac{18c \left( 4cr^8 - 10cr^6 - 6r^5 - 18r^2 + 9 \right)}{\left( 2cr^6 + 9 \right)^2} .
$$

For $r = \epsilon$ where $\epsilon$ is sufficiently small, $\frac{\mathrm{d}\kappa(r, c_t)}{\mathrm{d}r} > 0$ because the numerator is positive and the denominator is positive:

$$
\lim_{\epsilon \to 0^+} 4cr^8 - 10cr^6 - 6r^5 - 18r^2 + 9 = 9 > 0 .
$$

However, when $r = 1$, the numerator becomes negative and as such $\frac{\mathrm{d}\kappa(r, c_t)}{\mathrm{d}r} < 0$:

$$
18c \left( 4c - 10c - 6 - 18 + 9 \right) = 18c \left( -6c - 15 \right) < 0 .
$$

Since $\frac{\mathrm{d}\kappa(r, c_t)}{\mathrm{d}r}$ is continuous over $[\epsilon, 1]$, the switch of signs between the endpoints shows that there exists an interior point where $\frac{\mathrm{d}\kappa(r, c_t)}{\mathrm{d}r} = 0$. $\qquad \square$

**Lemma A.3.** *Let $r_1$ and $c_1$ be the values guaranteed to exist by Lemma A.2 such that $\frac{\mathrm{d}\kappa(r_1, c_1)}{\mathrm{d}r_1} = 0$. For $c_2 < c_1$, we have*

$$
\frac{\mathrm{d}\kappa(r_1, c_2)}{\mathrm{d}r_1} > 0 .
$$

*Proof.* Since $r_1$ is the radius that maximizes mutual information at $c_1$ (which we know to exist based on Lemma A.1), we know that $\kappa(r_1, c_1) > \kappa(r, c_1) \quad \forall\, 0 < r \leq 1$. Further, with

$$
\frac{\mathrm{d}\kappa}{\mathrm{d}r} = \frac{18c \left( 4cr^8 - 10cr^6 - 6r^5 - 18r^2 + 9 \right)}{\left( 2cr^6 + 9 \right)^2}
$$

where $\frac{\mathrm{d}\kappa(r_1, c_1)}{\mathrm{d}r_1} = 0$, we focus on only the numerator:

$$
\begin{aligned}
4c_1 r_1^8 - 10c_1 r_1^6 - 6r_1^5 - 18r_1^2 + 9 &= 0 \\
4c_1 r_1^8 - 10c_1 r_1^6 &= 6r_1^5 + 18r_1^2 - 9 .
\end{aligned}
$$

Suppose that $r_2 = r_1 + \delta$. Since we know that $\frac{\mathrm{d}\kappa(r_1,c_1)}{\mathrm{d}r_1} = 0$ is a maximum, we need to show that $\frac{\mathrm{d}\kappa(r_1,c_2)}{\mathrm{d}r_1} > 0$. Since the denominator of $\frac{\mathrm{d}\kappa}{\mathrm{d}r_1}$ is strictly positive, we only need to show when the numerator is always positive

$$= 18c_2 \left(4c_2 r_1{}^8 - 10c_2 r_1{}^6 - 6r_1{}^5 - 18r_1{}^2 + 9\right) .$$

Substituting our equality from earlier,

$$= 18c_2 \left(4c_2 r_1{}^8 - 10c_2 r_1{}^6 - 4c_1 r_1{}^8 - 10c_1 r_1{}^6\right)$$
$$= 18c_2 r_1^6 (c_2 - c_1)(4r_1{}^2 - 10) .$$

Since $c_2 - c_1 < 0$, we need to see when $4r_1{}^2 - 10 < 0$ to maintain that the entire numerator is positive. This equality is true when

$$r_1 < \sqrt{\frac{10}{4}} ,$$

which is always true since $0 < r \le 1$. As such, we have established that $\frac{\mathrm{d}\kappa(r_1,c_2)}{\mathrm{d}r_1} > 0$. □

**Theorem 3.2.** *Assume that $\rho(\eta^{(i)}, \eta^{(j)}) = 1 - (\eta^{(i)} - \eta^{(j)})^2$ where $0 < r \le 1$. Let $r_1$ be the radius that maximizes mutual information for a given signal-to-noise ratio $c_1$. If we decrease the signal-to-noise ratio $c_2 < c_1$, then there exists $r_2 > r_1$ such that*

$$I(x_1^{(i)}, Y_{c_2}^{(r_2)}) > I(x_1^{(i)}, Y_{c_2}^{(r_1)}) .$$

*Proof.* Because $\frac{1}{2} \log$ is strictly increasing, any maximizer for $\kappa$ is also a maximizer for $I$.

As such, with Lemma A.3, we know that $\frac{\mathrm{d}\kappa(r_1,c_2)}{\mathrm{d}r_1} > 0$. Therefore, $r_2 = r_1 + \delta$ would increase mutual information, proving that we need to increase $r$ beyond $r_1$ to reach the maximum mutual information at SNR level $c_2$. □

### A.2. Noising over Positions

**Proposition 3.3.** *Let $\|\eta_1^{(i)} - \eta_1^{(j)}\|_2 = \gamma << 1$. Given variance-preserving noise and correlation structure between positions as $\rho$, the expected squared distance between node $i$ and $j$ with respect to noise level $t$ is*

$$\mathbb{E}[(\eta_t^{(i)} - \eta_t^{(j)})^2] = 2(1 - t) + 2\rho(\gamma)(1 - t) + t\gamma^2 .$$

*Proof.* Let $\eta_1^{(i)} = 0$ and $\eta_1^{(j)} = \gamma$ without loss of generality. By definition of variance-preserving noise, we have

$$\eta_t^{(i)} \sim \mathcal{N}(0, 1 - t), \quad \eta_t^{(j)} \sim \mathcal{N}(\gamma, 1 - t) .$$

We then calculate for

$$\mathbb{E}[(\eta_t^{(i)} - \eta_t^{(j)})^2] = \mathrm{var}(\eta_t^{(i)} - \eta_t^{(j)}) + \mathbb{E}[\eta_t^{(i)} - \eta_t^{(j)}]^2$$

where

$$\mathrm{var}(\eta_t^{(i)} - \eta_t^{(j)}) = \mathrm{var}(\eta_t^{(i)}) + \mathrm{var}(\eta_t^{(j)}) + 2\,\mathrm{cov}(\eta_t^{(i)}, \eta_t^{(j)}) = 2(1 - t) + 2(1 - t)\rho(\gamma) ,$$

which results in

$$\mathbb{E}[(\eta_t^{(i)} - \eta_t^{(j)})^2] = 2(1 - t) + 2(1 - t)\rho(\gamma) + \gamma^2 \qquad \square .$$

## B. Experiment 1 & 2 Details

### B.1. Architecture

#### B.1.1. DYNAMIC MESSAGE PASSING

The model architecture used in the two experiment follows the same operations listed in Algorithm 1. The sole difference between the implementations of the models in the two experiments is the input and output node values. The remaining operations are identical between both experiments. We describe each operation in detail below.

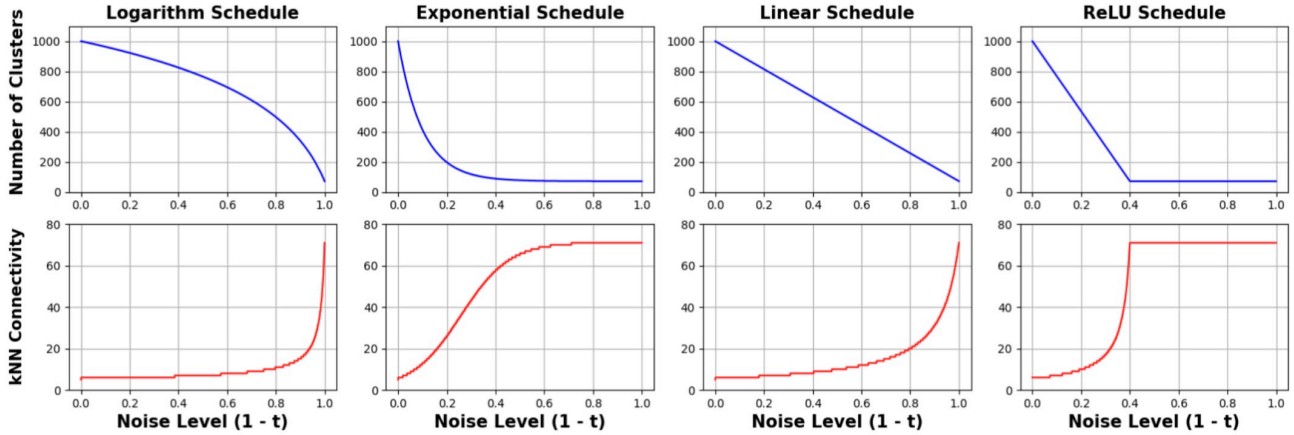

*Figure 5.* **Adaptive Schedules**. The schedule determines the level of coarse-graining and connectivity of the graph neural network corresponding to a noise level. (**Top**) The number of clusters (aka fewer clusters means higher coarse-graining). The higher the noise, the fewer number of clusters. (**Bottom**) The level of connectivity (i.e. range) determined by the $k$ in $k$NN. The higher the noise, the greater the $k$.

**Node Input Construction**. We define two sets of nodes: (1) the original nodes and (2) the coarse-grained nodes. Let $H_{\text{in}} = \{h_{\text{in}}^{(i)} \in \mathbb{R}^d\}_{i=1}^N$ be the original node features to the model, where $N$ is the number of nodes and $d$ is the dimension of the input data. Each coarse-grained node feature and position is defined as the average of the original nodes belonging to the specific coarse-grained node. Specifically, let $\hat{H}_{\text{in}} = \{\hat{h}_{\text{in}}^{(i)}\}_{i=1}^s$ and $\hat{R} = \{\hat{\eta}^{(i)}\}_{i=1}^s$ be the coarse-grained node features and positions, respectively, where $s$ is the number of clusters determined by adaptive schedule (Figure 5) at the current noise level. We determine which original node belongs to which coarse-grained node using voxel clustering. For each dimension $j$ of the node positions, we uniformly split the interval (with range $[\min(R_j), \max(R_j)]$) into $\sqrt[d]{s}$ partitions, which determines the boundaries of each cluster in the specific dimension $j$. If the position of an original node is within the boundaries of a cluster (for all dimensions), it belongs to that cluster.

For ModelNet40, the initial node input $h_{\text{in}}^{(i)} = \text{concat}(x^{(i)} = \emptyset, \eta^{(i)}, t)$ where $\eta \in \mathbb{R}^3$ and $t \in [0, 1]$. Since ModelNet40 only has positions and no features of each node, $x$ is set to be a no-op; hence, we have dimensions of $d = 4$ for node feature $h_{\text{in}}^{(i)}$.

The spatiotemporal transcriptomic dataset follows a similar form, except now each node (i.e. cell) has both positional and feature information (where features are the gene expressions). Thus, the initial input (as described in Section 5.2) is $h_{\text{in}}^{(i)} = \text{concat}(x^{(i)}, \eta^{(i)}, t)$ where $x \in \mathbb{R}^3$ since we are tracking 3 genes, $\eta \in \mathbb{R}^2$ since we concatenate the 1-dimensional temporal axis (progression of cell through time) and the 1-dimension spatial axis (position of cell across a line), and $t \in [0, 1]$ is the noising time of the generative model (not to be confused with physical time of cells). Therefore, we have dimensions of $d = 6$ for node feature $h_{\text{in}}^{(i)}$.

**Edge Construction**. Given the current noise level $t$, we leverage the adaptive schedule (Figure 5) to provide the $k$NN connectivity between the coarse-grained nodes. Specifically, we define our edges as

$$\mathcal{E} = k\text{NN-Graph}(\hat{R}) \ .$$

**Lifting Operation**. Before message passing, we first *lift* or embed each initial node input (both original and coarse-grained nodes) with

$$H_{k=0} = \text{MLP}(H_{\text{in}}; [d, \text{hdim}, \text{hdim}, \text{hdim}]) \ .$$

where hdim is the hidden dimensions and the original and coarse-grained nodes have separately parameterized MLPs. Note that $k = 0$ refers to the node features before the first message passing layer and is not related to the $k$NN connectivity.

**Coarsening**. For each layer in our model (the layer index is denoted as $k$), we define the message-passing scheme from the original nodes to the coarse-grained nodes as the following:

$$\hat{h}_k^{(i)} = \sum_{j \in \mathcal{N}_{\text{cluster}}(i)} \text{MLP}_k\Big(\Big\{\text{linear}_k\big(\text{concat}\{\hat{h}_k^{(i)}, h_k^{(j)}\}\big), \text{linear}_k(\hat{\eta}^{(i)} - \eta^{(j)}), \text{linear}_k(\mathbf{e}_{ij})\Big\}; [3\text{hdim}, \text{hdim}, \text{hdim}]\Big)$$

where $\mathbf{e}_{ij} = \|\hat{\eta}^{(i)} - \eta^{(j)}\|_2$. We conduct this operation for all original nodes $j$ in the cluster $i$ (denoted as $\mathcal{N}_{\text{cluster}}(i)$). All linear layers map to $\mathbb{R}^{\text{hdim}}$.

**Message Passing Layer**. After the coarsening operation, message passing is conducted between the coarse-grained nodes. For each message passing layer $\text{MP}_k$, we have the following

$$\hat{H}_k = \text{MP}_k(\hat{H}_k, \mathcal{E}) \quad \text{where} \quad \text{MP}_k \in [\text{GCNConv}, \text{GATConv}].$$

For every experiment, we evaluate both GCNConv and GATConv as the message passing schemes to demonstrate that the performance benefit of DMP generalizes to more than just one type of message passing. We define GCNConv and GATConv following (Fey & Lenssen, 2019) for each node $i$ as

$$\text{GCNConv}(\hat{h}_k^{(i)}, \mathcal{E}^{(i)}) = \text{linear}\left(\sum_{j \in \mathcal{E}^{(i)}} \frac{1}{S} \hat{h}_k^{(j)}\right), \quad \text{GATConv}(\hat{h}_k^{(i)}, \mathcal{E}^{(i)}) = \alpha_{i,i}\text{linear}_s(\hat{h}_k^{(i)}) + \sum_{j \in \mathcal{E}_i} \alpha_{i,j}\text{linear}_t(\hat{h}_k^{(j)})$$

where $\mathcal{E}^{(i)}$ give the nodes incident to node $i$. Further, $\alpha_{i,j}$ is calculated as

$$\alpha_{i,j} = \frac{\exp(\text{LeakyReLU}(\mathbf{a}_s^\top \text{linear}(\hat{h}_i) + \mathbf{a}_l^\top \text{linear}(\hat{h}_k^{(j)})))}{\sum_{r \in \mathcal{E}^{(i)}} \exp(\text{LeakyReLU}(\mathbf{a}_s^\top \text{linear}(\hat{h}_k^{(i)}) + \mathbf{a}_l^\top \text{linear}(\hat{h}_k^{(r)})))}$$

where $\mathbf{a}_s$ and $\mathbf{a}_t$ are learnable source and target attention vectors.

**Uncoarsening**. Similar to the coarsening operation, we now message pass from the coarse-grained nodes back to the original nodes since we ultimately want to predict the velocity field with respect to the original nodes:

$$\text{spread}^{(i)} = \text{MLP}_k\left(\left\{\text{linear}_k\left(\text{concat}\{h_k^{(i)}, \hat{h}_k^{(j)}\}\right), \text{linear}_k(\eta^{(i)} - \hat{\eta}^{(j)}), \text{linear}_k(\mathbf{e}_{ij})\right\}; [3\text{hdim}, \text{hdim}, \text{hdim}]\right)$$

where $\mathbf{e}_{ij} = \|\eta_i - \hat{\eta}_j\|_2$ and $j$ is the coarse-grained node that the original node $i$ belongs to.

Then, we use a gating mechanism $\lambda$ to control the flow of information from the coarse-grained nodes to the original nodes:

$$H_{k+1} = \text{MLP}_k(\lambda H_k + (1 - \lambda)\text{spread}; [\text{hdim}, \text{hdim}, \text{hdim}]]), \quad \text{where} \quad \lambda = \text{linear}(\text{concat}(H_k, \text{spread})).$$

**Project Operation**. Finally, after conducting the past operations over $k \in K$ layers, the node features $H_{k+1}$ are projected back to the ambient space:

$$v_\theta^{(\text{pred})} = H_{\text{out}} = \text{MLP}(H_K; [\text{hdim}, \text{hdim}, \text{hdim}, \text{odim}])$$

where odim is the dimension of the output. Depending on the task (i.e. whether we are generating positions or generating features), odim can either be the dimensions of the positions or the dimensions of the features. Specifically, for ModelNet40, since we are generating positions, odim $= 3$ ($xyz$ coordinates), and for the spatial transcriptomics dataset, since we are generating features, odim $= 3$ (number of genes).

### B.1.2. BASELINES

To ensure a fair comparison between baseline methods and DMP, we use the same architecture from Appendix B.1.1, except the message passing edges are fixed (kNN, fully connected, or long-short range) and the graph is always at full resolution. Specifically, the baselines consider the original nodes as the coarse-grained nodes ("full-resolution") and conduct the coarsening and uncoarsening operations towards these "full-resolution" nodes (i.e. one-to-one mapping rather than DMP's many-to-one and one-to-many mappings). This consistency between baselines and DMP is important since it shows that the performance improvement purely comes from the adaptive range and resolution of the graph rather than architectural improvements.

### B.2. Hyperparameters

We use the hyperparameter values listed Table 5 for both experiments. Many of these values are borrowed from the default values in Tong et al. (2023). Additionally, it was shown in previous work (Tong et al., 2023) that flow-matching requires fewer NFEs than diffusion-based models, which is why we use more NFEs for diffusion compared to flow-matching.

| Parameter | Value | Description |
|---|---|---|
| Activation | GELU | Neural network activation function |
| Normalization | Batch Norm | Layer normalization type |
| Message Passing Layers | 3 | Number of message passing layers |
| Hidden Dimensions | 64/32 | Dimension of hidden features (Exp. 1/Exp. 2) |
| Training Epochs | 300 | Total number of training epochs |
| Learning Rate | 0.0001/0.001 | Initial learning rate (Exp. 1/Exp. 2) |
| Scheduler | LambdaLR | Learning rate scheduler type |
| Scheduler Warmup | 10 | Number of warmup epochs |
| EMA Decay | 0.95 | Exponential moving average decay rate |
| Batch Size | 128 | Number of samples per batch |
| kNN (t=1) | Fully connected | k-nearest neighbors at time t=1 |
| kNN (t=0) | $c = \lceil \sqrt[3]{N} \rceil$ | k-nearest neighbors at time t=0 |
| Clusters (t=1) | $\lceil \sqrt{cN} \rceil$ | Number of coarse-grained nodes at t=1 |
| Clusters (t=0) | $N$ | Number of coarse-grained nodes at t=0 |
| NFEs (Diffusion) | 1000 | Number of function evaluations for diffusion |
| NFEs (Flow-Matching) | 200 | Number of function evaluations for flow-matching |

*Table 5.* Hyperparameter values and descriptions used for experiments 1 & 2.

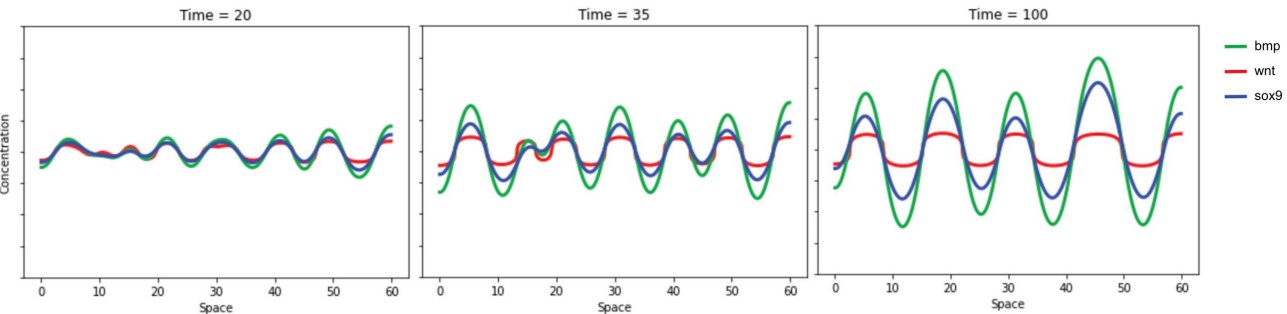

*Figure 6.* Example of the spatiotemporal data generated by the reaction-diffusion equation.

### B.3. Spatiotemporal Transcriptomics

**Dataset Construction**. Spatiotemporal transcriptomic measures the gene expression of many single-cells distributed spatially (e.g. a tissue slice) over multiple timepoints. A graph in this context can be viewed as having N nodes (each node is a cell) where the features represent the expression levels of three genes (Bmp, Sox9, and Wnt). The position of each node combines both temporal and spatial information; specifically, a one-dimensional temporal coordinate indicating the progression of the cell through time, and a one-dimensional spatial coordinate representing the cell's position along a line across the developing hand tissue. These two coordinates together form the complete positional information for each cell.

This experiment aims to generate the distribution over features while holding the positions fixed. We simulate the trajectory of spatial transcriptomic following Raspopovic et al. (2014). Each gene expression value over time and space is determined by the following reaction-diffusion equation:

$$\frac{\partial \, \text{sox9}}{\partial t} = \alpha_{\text{sox9}} + k_2 \, \text{bmp} - k_3 \, \text{wnt} - \text{sox9}^3$$

$$\frac{\partial \, \text{bmp}}{\partial t} = \alpha_{\text{bmp}} - k_4 \, \text{sox9} - k_5 \, \text{bmp} + d_b \nabla^2 \, \text{bmp}$$

$$\frac{\partial \, \text{wnt}}{\partial t} = \alpha_{\text{wnt}} - k_7 \, \text{sox9} - k_9 \, \text{wnt} + d_w \nabla^2 \, \text{wnt}$$

where $\alpha \in \mathbb{R}^l$ represents the initial concentration for each gene, $k$ represents the regulation coefficient of one gene on

another gene, and $\mathrm{sox9}, \mathrm{bmp}, \mathrm{wnt} \in \mathbb{R}^l$ represents the gene concentration at the current time point. $l$ is the number of spatial values (the higher the value, the larger the space) where each spatial value is a scalar. The other variables and their specific values are further detailed in Table 6.

For our training set, we generate a dataset of 10000 samples by simulating the reaction-diffusion equation under different initial conditions. For each simulation, we randomly initialize the concentration of each gene by sampling from $\mathcal{U}(-0.01, 0.01)$. An example of the generated data is shown in Figure 6. We then divide the spatial dimension into 10 equally spaced points and record the system's evolution at 10 different timepoints, resulting in graphs with 100 nodes (10 spatial points $\times$ 10 timepoints).

Since graph neural networks are discretization invariant (Li et al., 2020), we generate a test set of 2000 samples using a different spatial and temporal resolution. Specifically, we use 8 spatial points and 12 timepoints, resulting in graphs with 96 nodes.

We detail the values for variables used to simulate the reaction-diffusion equation in Table 6, which are directly taken from Raspopovic et al. (2014). Further, we can visualize the reaction-diffusion equation with a Turing system displayed in Figure 7.

**Biologically-Inspired Tasks**. We also visualize the conditional tasks that we evaluate DMP and baselines (Section 5.2) on in Figure 8. The details and biological relevance of each task is as follows:

1. (**Temporal Trajectory Prediction**). Given the gene expression values over space for the first timepoint, predict the future time trajectory of the gene expression values over space. This is often the default task of current spatiotemporal transcriptomics models.

2. (**Temporal Interpolation**). Predict the trajectories that begin at the given starting spatial gene expressions and terminate at the given ending spatial gene expressions. For example, we may want to find the path skin cells (starting timepoint) take to reprogram into stem cells (ending timepoint).

3. (**Gene Imputation**). Given a missing gene over space and timepoints, predict the likely expression of this gene conditioned on the other genes. For example, some instruments can only measure a subset of genes and we may want to impute the expressions of the other genes not in the subset.

4. (**Spatial Imputation**). With a missing spatial portion, impute the expressions of this missing spatial portion. For example, many spatial transcriptomics measurements may have portions of the tissue missing (due to experimental error). We may want to impute this missing portion of the tissue.

5. (**Gene Knockout**). If we knockout a gene, predict what the other gene expressions will look like. For example, we may be interested in how a gene contributes to disease; by knocking out this gene, what are the resulting spatial and temporal gene expressions?

**Importance**. Having the ability to model these types of tasks and answer biologically relevant questions—that previous models (Peng et al., 2024; Qiu et al., 2024) could not—is incredibly significant. By developing an expressive yet efficient flow-based generative model, we take a first step towards a foundation model for modeling spatiotemporal data, particularly for single-cell transcriptomics.

## C. Experiment 3 Details

Here, we elaborate on the ImageNet experiment and the details of modifying the Diffusion Transformer (DiT) to DMP.

### C.1. Modifying a Baseline SOTA Model to DMP

**Baseline**. For a given $256 \times 256$ pixel image from ImageNet, the baseline DiT model first encodes the image to $32 \times 32$ using the encoder of a pretrained autoencoder. Since it is computationally infeasible to employ full (quadratic) attention over all $32 \times 32 = 1024$ elements, the model *patchifies* the $32 \times 32$ sample into coarser versions based on a chosen patch size $p \in \{2, 4, 8\}$. The larger the patch size, the coarser the representation of the image. For example, $p = 2$ results in a $16 \times 16$ representation while $p = 8$ results in a $4 \times 4$ representation. The model then uses a transformer to fully attend to these

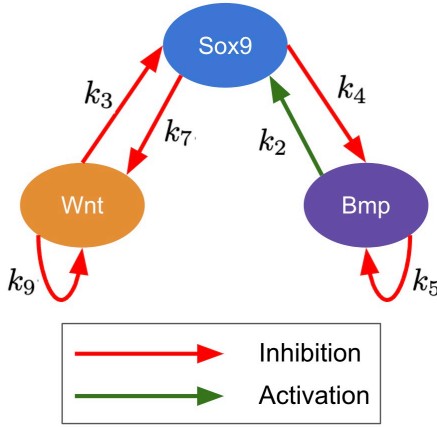

*Figure 7.* **Turing Network for Bmp-Sox9-Wnt** (Raspopovic et al., 2014).

| Parameter | Value | Description |
|---|---|---|
| $\alpha_{\text{sox9}}$ | $\alpha \sim \mathcal{U}(-0.01, 0.01)$ | Base production rate of Sox9 protein |
| $\alpha_{\text{bmp}}$ | $\alpha \sim \mathcal{U}(-0.01, 0.01)$ | Base production rate of BMP |
| $\alpha_{\text{wnt}}$ | $\alpha \sim \mathcal{U}(-0.01, 0.01)$ | Base production rate of WNT |
| $k_2$ | 1 | Positive regulation coefficient of BMP on Sox9 |
| $k_3$ | $-1$ | Negative regulation coefficient of WNT on Sox9 |
| $k_4$ | 1.27 | Negative regulation coefficient of Sox9 on BMP |
| $k_5$ | $-0.1$ | Self-regulation coefficient of BMP |
| $k_7$ | 1.59 | Negative regulation coefficient of Sox9 on WNT |
| $k_9$ | $-0.1$ | Self-regulation coefficient of WNT |
| $d_b$ | 1 | Diffusion coefficient for BMP |
| $d_w$ | 2.5 | Diffusion coefficient for WNT |

*Table 6.* Parameter values and descriptions for the reaction-diffusion system that generated the spatiotemporal transcriptomics dataset in Section 5.2.

patches to learn how to reverse the noise. We refer readers to Peebles & Xie (2023) for details on the image transformations (moving image to latent space, patching, etc) and the transformer architecture used within this model.

We set the patch size for the baseline DiT to be $4$ because it enables fair comparison with DiT-DMP while allowing DiT-DMP to use multiple patch size configurations—with $p = 4$, DiT-DMP can adaptively utilize patch sizes of $1$, $2$, and $4$, whereas choosing a smaller baseline patch size would constrain DiT-DMP's range of patch sizes that it can use. Further, we leverage on the Small model (Peebles & Xie, 2023) because while larger models like Large or X-Large could potentially achieve better absolute performance, our primary goal is to demonstrate that DiT-DMP can effectively adapt a state-of-the-art architecture to improve relative performance, which can be shown equally well with the Small model.

**DiT-DMP**. To modify DiT to DiT-DMP, we condition (1) $p$ on the noise (i.e. $p_t$) rather than being a fixed value (i.e. $p \in \{2, 4, 8\}$) and (2) the range of attention on the noise rather than being fixed to full or quadratic attention. Like the two previous experiments, we leverage the exponential adaptive schedule to determine the level of coarse-graining (aka patching) and the range of attention. However, because the patch size must be a divisor of $32$, we must round the coarse-graining value given by the exponential schedule to the nearest divisor value.

Respecting the notation established in Section 4.1, let $s_{t \in [0,1]}$ be the length of a side of the patchified representation (for instance, $s_0 = 8$ means that our patchified representation is $8 \times 8$ and our patch size is $p = 4$) at time $t$. We choose resolution endpoints $s_0 = 8$ (resolution at complete noise) and $s_1 = 32$ (resolution at no noise). Correspondingly, we choose range endpoints $r_0$ and $r_1$—the kernel size of the attention at $t = 0$ and $t = 1$—to be such that the computational complexity matches the baseline DiT of some given patch size with quadratic complexity. Intermediate $s_t$ and $r_t$ values are the exponential interpolation between the endpoints rounded to the nearest divisor. Specifically, for some $t$, we have $(r_t, s_t) \in \{(8, 8), (4, 16), (2, 32)\}$.

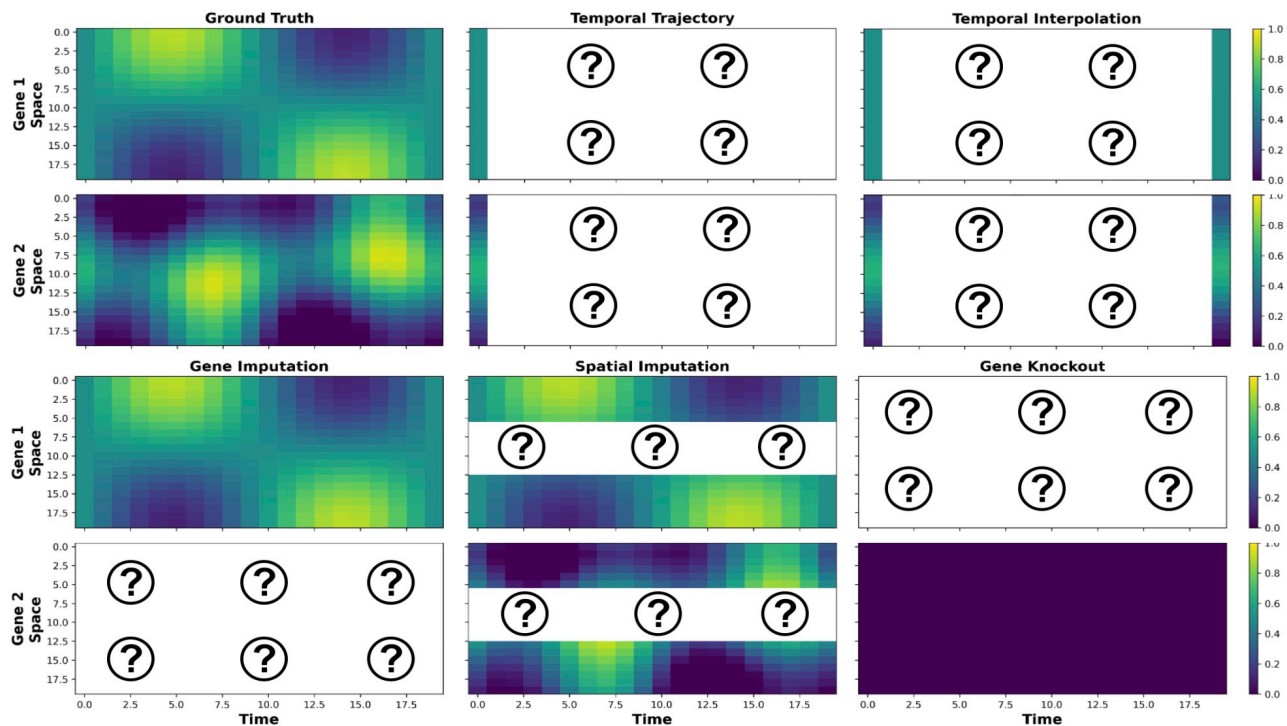

*Figure 8.* **Spatiotemporal Transcriptomic Tasks**. In addition to unconditional generation, these are several other useful *conditional* tasks for single cell transcriptomics. For each task, we delineate what the model is given as conditioning information and what the model needs to generate with "?".

Notice that unlike DiT, which is limited to $p \in \{2, 4, 8\}$ due to the computational complexity of quadratic attention, DiT-DMP can have $p = 1$ at $t = 1$ (aka full resolution) since we adapt the attention to be of a smaller range than full attention.

### C.2. Hyperparameters

| Parameter | Value |
|---|---|
| Layers | 12 |
| Hidden size | 384 |
| Heads | 6 |
| Batch size | 64 |
| Number of iterations | 800K |
| NFEs | 250 |
| Global Seed | 0 |

*Table 7.* Hyperparameter Values for DiT (and the DMP variant).

We list all hyperparameters used for this experiment in Table 7. All hyperparameters used in this experiment are the same as Peebles & Xie (2023) except batch size and number of training iterations. Rather than a batch size of 256 and 400K training iterations, we have a batch size of 64 with 800K training iterations due to compute limitations.

## C.3. Examples of Generated Images

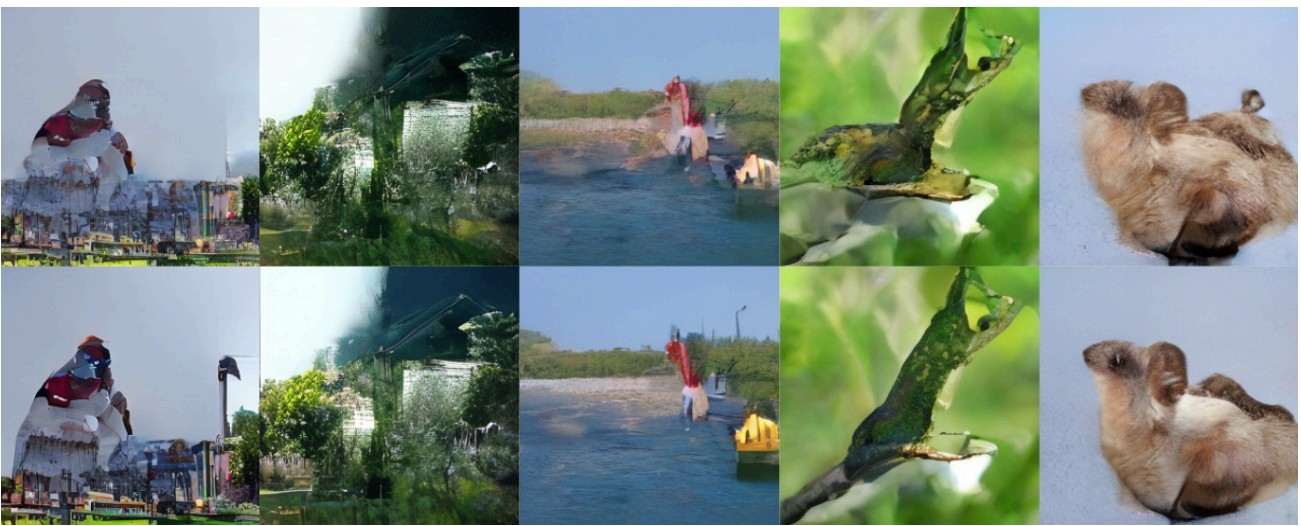

*Figure 9.* Select predictions from (**Top Row**) DiT and (**Bottom Row**) DiT-DMP.

# D. Additional Experiments & Ablations

## D.1. Empirical Analysis: Resolution

Using the same dataset from Section 3.2, we also visualize the average Gromov-Wasserstein (GW) distance between the geometric graphs and progressively noised and coarse-grained versions of the geometric graphs. Specifically, for each geometric graph, we generate increasingly noisy versions of the graph (isotropic noise added to the positions). Then, for each noised version of the graph, we coarse grain the graph by either max or mean pool the node positions. We then calculate the GW distance between the original graph and the coarse-grained noisy version.

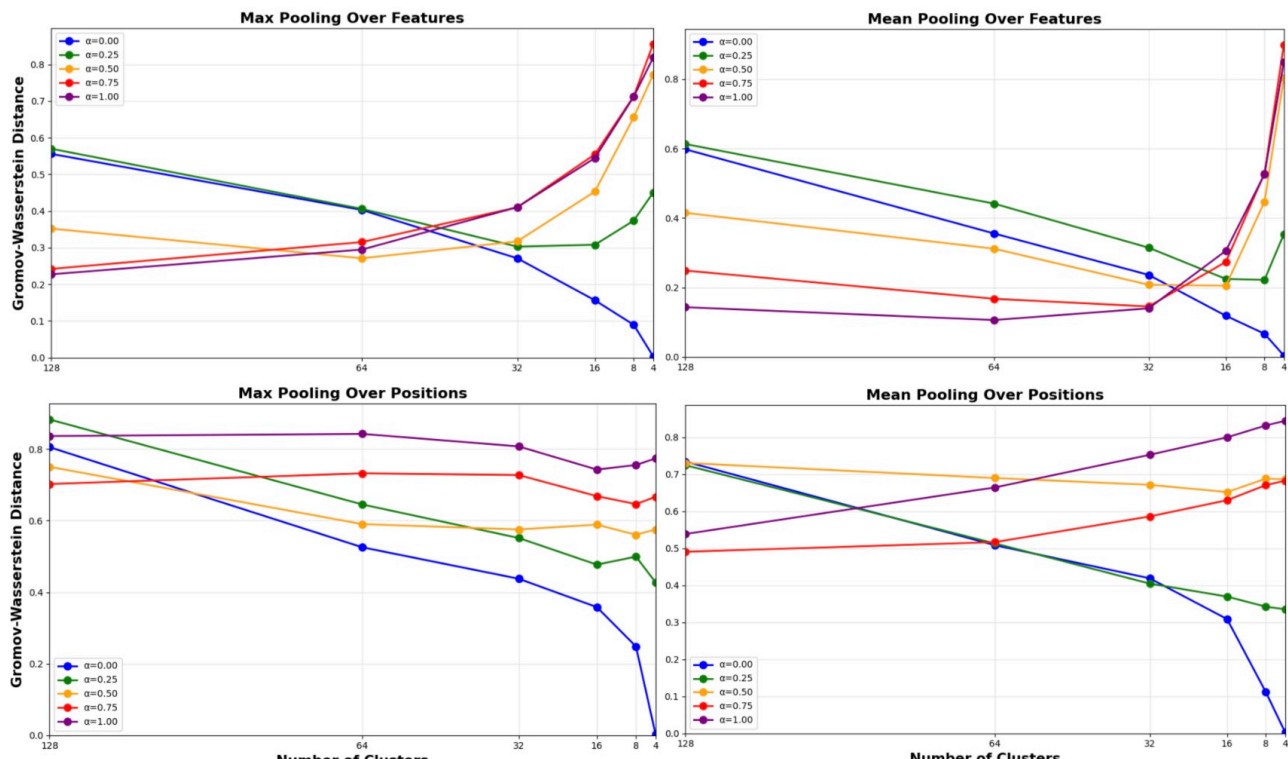

*Figure 10.* Gromov-Wasserstein distance between the original unnoised geometric graph and coarse-grained versions of the geometric graph under different levels of noise ($\alpha = 1 - t$, meaning higher $\alpha \rightarrow$ higher noise). Two types of coarse-graining (max and mean pooling) are conducted over either positions or features.

**Results**. Figure 10 summarizes the results of this analysis. The key pattern to note is that as we increase the level of coarse-graining (i.e. lower the number of clusters), low-noise versions of the graph increase in GW distance drastically while high-noise versions decrease in GW distance as we coarse-grain. This decrease in GW distance for high-noise versions makes sense: as we increase noise, the coarse-graining effectively smooths out the noise, better preserving the original signal. This finding is also supported by Theorem 3.2, which shows that increasing the radius of aggregation (e.g. level of coarse-graining where the aggregation scheme is the mean or max) is necessary to maximize mutual information for increasing noise levels.

Figure 4 is derived from this experiment where we instead visualize the number of clusters that minimize the GW distance for each noise level.

### D.2. Performance with Increasing Number of Layers

There are three popular issues related to the number of layers in a graph neural network and performance: (1) oversmoothing (Oono & Suzuki, 2020), where node representations become too similar as layers increase; (2) underreaching (Alon & Yahav, 2021), where the network has too few layers to capture long-range dependencies, preventing information from propagating between distant nodes that are relevant to the learning task; and (3) over-squashing (Alon & Yahav, 2021), where exponentially growing neighborhoods force the network to compress too much information into fixed-size vectors, leading to information bottlenecks in the message passing process.

Commonly, fully-connected graphs face issues of oversmoothing (since each node feature is the aggregation of all the node features in the graph) while fixed range graphs face issues of underreaching and over-squashing. However, since DMP at high noise levels is fully connected while at low noise levels are sparsely connected (we effectively interpolate based on the noise level between these two extremes), we investigate whether DMP can resolve these common GNN-related issues by leveraging the benefits of both kinds of connectivity (i.e. both dense and sparse connectivity).

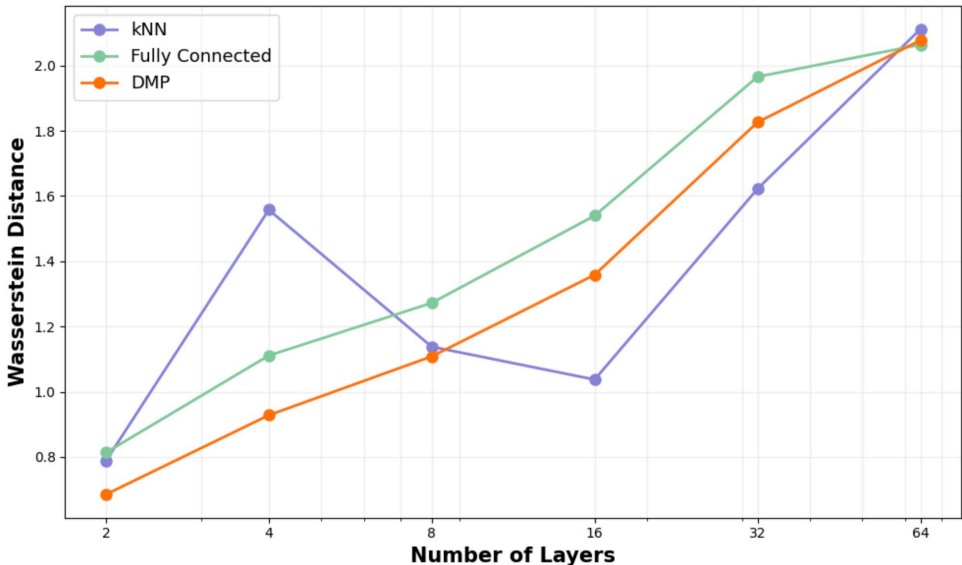

*Figure 11.* Performance of models on the spatiotemporal transcriptomics dataset across an increasing number of message passing layers. We report the Wasserstein distance for the unconditional generation task for DMP, a kNN graph, and a fully connected graph.

We train our method and baselines (kNN and fully connected) using the same architecture as described in Appendix B.1 on the spatiotemporal transcriptomics dataset, except with a hidden dimension of 32 due to computation limitations. Critically, we vary the number of message passing layers $K$ and evaluate the performance of each method on the unconditional generation task with a different number of layers $K$. We use GCNConv as the message passing architecture.

**Results**. Figure 11 plots the results of this experiment. At the few number of layers regime ($2 - 4$ layers), DMP dominates in performance while kNN performs significantly worse compared to fully connected. This finding is expected because, without many layers, kNN faces the underreaching problem since far away nodes cannot exchange information; the fully connected baseline does not have this issue. However, at the higher number of layers regime ($8+$ layers), we see that the Wasserstein distance of both DMP and fully connected rises steadily while kNN decreases its Wasserstein distance between $8 - 16$ layers before increasing again. Eventually, at the highest number of layers ($64$ layers), all methods trend towards effectively random predictions (since random predictions result in a wasserstein distance of $2.449$ for the unconditional generation task as provided in Table 3).

Surprisingly, DMP faces similar issues as fully connected where increasing the number of layers directly decreases its performance. Such a result demonstrates that DMP still suffers from the issues of oversmoothing as fully connected baselines. Ultimately, the lowest Wasserstein distance is still achieved by DMP with fewer number of layers.

