# OpenReview forum: "Geometric Generative Modeling with Noise-Conditioned Graph Networks"
_ICML.cc/2025/Conference — ICML 2025 poster_

### Official Review · Reviewer_GxaG · 2025-03-13

**Overall Recommendation:** 3

**Summary:**

The paper introduces ​Noise-Conditioned Graph Networks (NCGNs), a class of graph neural networks that dynamically adapt their architecture based on the noise level during flow-based generative modeling of geometric graphs. The key innovation is ​Dynamic Message Passing (DMP), which adjusts both the connectivity range (e.g., k-nearest neighbors) and graph resolution (via coarse-graining) as a function of noise.

**Claims And Evidence:**

**Strengths:**
​
- It is the first work to formalize noise-dependent graph architecture adaptation for geometric generative modeling.
Integrates theoretical insights (mutual information analysis) with practical algorithmic design (DMP).

- Comprehensive Experiments: Validated on 3D shapes, biological data, and images, demonstrating broad applicability.
​Theoretical-Experimental Alignment: Attention weight analysis (Fig. 3) and Gromov-Wasserstein distance measurements (Fig. 4) empirically confirm theoretical predictions.

​**Weaknesses:**

- The ​scheduler for adjusting resolution/connectivity (Fig. 5) is predefined (e.g., exponential), not learned. This may limit optimality across diverse datasets.
- Coarsening via voxel clustering or K-means could introduce artifacts; learnable pooling (e.g., DiffPool) might improve performance.
​Baseline Comparisons:

**Essential References Not Discussed:**

-

**Experimental Designs Or Analyses:**

The experimental design seems to be valid.

**Methods And Evaluation Criteria:**

The method and Evaluation Criteria in this paper seems to be sound.

**Other Comments Or Suggestions:**

The paper presents a ​theoretically grounded and empirically validated method for noise-adaptive graph generation. While some assumptions simplify real-world complexity, the core claims are well-supported, and the approach demonstrates clear practical value. Future work could explore learned schedulers and broader baseline comparisons.

**Other Strengths And Weaknesses:**

-

**Questions For Authors:**

**Q1:** The paper uses predefined schedulers (e.g., exponential) to interpolate between connectivity and resolution boundaries. Have you explored learned schedulers (e.g., via meta-learning or reinforcement learning) to dynamically optimize $r_t$ and $s_t$ during training? If not, do you foresee such approaches improving performance, particularly in domains where noise dynamics are non-monotonic or dataset-specific?

**Q2:** Recent works like E(n)-GNNs and Equivariant Diffusion models explicitly encode geometric symmetries (e.g., rotation/translation invariance) into graph networks. How does DMP complement or contrast with these methods?  Is there a way to integrate equivariance constraints into NCGNs, is it possible that this would enhance performance on tasks like molecular generation, where symmetry preservation is critical?

**Relation To Broader Scientific Literature:**

-

**Theoretical Claims:**

Correlation function in Theorem 3.2 is idealized; real-world spatial correlations may be more complex.

---

> ### Author Rebuttal · Authors · 2025-03-27
>
> We thank the reviewer for their time, effort, and constructive feedback. We address the reviewer’s concerns and questions below:
>
> > Learned scheduler
>
> **Implementing learnable schedules is difficult** because the schedule is used for kNN graph construction and coarse-graining procedures, which involve non-differentiable operations (sorting, clustering, discrete assignments) that are not amenable to standard backpropagation. Instead of relying purely on heuristics, we conducted ablation studies in Table 2 and Section 5.1.1 comparing four scheduling functions (linear, exponential, logarithmic, ReLU), finding that all significantly outperform fixed baselines, with exponential consistently performing best.
>
> While developing differentiable approximations for these operations represents an exciting future direction, we believe it warrants dedicated study beyond our current paper's scope.
>
> > Coarsening via voxel clustering or K-means could introduce artifacts; learnable pooling (e.g., DiffPool) might improve performance
>
> We agree that learnable pooling methods like DiffPool could potentially enhance performance. We emphasize that NCGN is a general framework where specific implementation details (including coarsening operations) can be customized. We intentionally selected simple, deterministic pooling methods to clearly demonstrate our core contribution—the benefit of noise-conditioned architectures—without confounding factors from complex learnable components. Our experimental results show that even with these basic coarsening operations, NCGNs deliver significant performance improvements across multiple domains. Future work could certainly explore more advanced coarsening operations within the NCGN framework.
>
> > E(n)-GNNs and Equivariant Diffusion models explicitly encode geometric symmetries [...]. Is there a way to integrate equivariance constraints into NCGNs?
>
> **DMP complements these equivariant methods**. As a general framework, NCGNs can incorporate any type of GNN layer, including equivariant ones. For instance, DMP could be implemented with equivariant layers like those in E(n)-GNNs, SchNet, or DimeNet that incorporate pairwise distances between nodes. Our theoretical analysis in Section 3 could also inform best practices for equivariant architectures—for example, many equivariant models operate on radius graphs, and our work provides principled guidance on how the radius should vary with noise level.
>
> While further investigation is needed, we believe the performance gains from NCGNs could potentially translate to equivariant models as well, making the application of NCGNs in structural biology and other geometric domains a promising direction for future research.
>
> > Correlation function in Theorem 3.2 is idealized; real-world spatial correlations may be more complex.
>
> We agree that the correlation function used in Theorem 3.2 is a simplified model of spatial correlation. This simplification was necessary to make theoretical analysis tractable while still capturing the essential property that correlation between nodes decreases with distance. Importantly, we designed our work to not rely solely on theoretical guarantees from idealized assumptions. Section 3.2 provides empirical analysis on real 3D geometric objects, demonstrating that our key insights about optimal reception fields and resolution scaling with noise also hold true in practice.
>
> > Broader baseline comparisons
>
> Our primary contribution is the noise-conditioning framework of NCGNs that can enhance existing architectures including many SOTA approaches. We implement NCGN on top of widely used architectures like GCNs and GATs and compare NCGN against general connectivity patterns including sparse connections (kNN), long and short connections, and full (quadratic) connections. In fact, **these effectively generalize many SOTA approaches**: the fully-connected GAT baseline is effectively a graph transformer and the long-short range baselines capture similar connectivity patterns as architectures with virtual nodes. Furthermore, Section 5.3 demonstrates that incorporating noise-conditioning to a SOTA transformer-based model results in significant performance gains.
>
> ---
> Thank you again for your helpful feedback. We welcome any additional discussions.

---

### Official Review · Reviewer_qfiH · 2025-03-14

**Overall Recommendation:** 4

**Summary:**

This paper introduces Noise-Conditioned Graph Networks (NCGNs), a generative modeling approach for geometric graphs that dynamically adjusts graph structure based on noise levels rather than keeping it fixed throughout the process. The authors propose a method to adapt how information flows through the graph depending on the noise intensity.

At the core of this approach is Dynamic Message Passing (DMP), which modifies both the connectivity range (how far messages travel) and the resolution (level of detail in the graph representation) as noise changes. When noise is high, the model broadens connections and simplifies the graph structure; as noise decreases, it shifts to a more refined representation. The authors support this design with theoretical insights, showing that stronger noise necessitates information aggregation from more distant nodes and that coarser graph structures can reduce complexity while preserving essential information.

To validate the approach, the authors evaluate DMP on 3D shape generation, spatial transcriptomics, and image generation, demonstrating that it consistently outperforms existing graph-based generative models. Importantly, the method maintains linear-time complexity, making it scalable for large datasets.

This work highlights the importance of structured noise in generative models, arguing that their graph representations should evolve accordingly. The results suggest that noise-adaptive architectures lead to more expressive and efficient generative models across different domains.

**Claims And Evidence:**

Most claims are well-supported by theory and experiments, but some areas need further validation. The theoretical analysis and empirical results strongly support the idea that noise-adaptive message passing improves generative modeling. Experiments across 3D shape generation, spatial transcriptomics, and image generation show DMP consistently outperforms fixed architectures, with clear improvements in Wasserstein distance. The paper also convincingly explains how its scheduling strategy ensures linear-time complexity.

However, some claims require more justification. The theoretical analysis assumes isotropic Gaussian noise and continuous graphs, which may not hold in all real-world settings. More validation on structured or irregular noise would strengthen this claim. The choice of an exponential scheduling function is somewhat heuristic, and a deeper study of learned schedules could provide stronger justification. While the method performs well on tested datasets, its applicability to other domains like molecular modeling or social networks remains uncertain. A discussion of these limitations would improve clarity. This I'll get back to later.

Overall, the paper presents strong evidence for its approach, but addressing theoretical assumptions, adaptive scheduling choices, and broader applicability would further strengthen the claims.

**Essential References Not Discussed:**

- The paper should discuss adaptive GNNs (e.g., Li et al., 2018; Shirzad et al., 2022) as related approaches for dynamically adjusting graph connectivity.
- Noise-aware graph learning (e.g., Luo et al., 2021; Zhang et al., 2020) is conceptually close to NCGNs and could be cited.
- Hierarchical graph pooling methods (e.g., Ying et al., 2018; Defferrard et al., 2019) share similarities with DMP's coarse-graining approach.
- Multi-resolution diffusion models (e.g., Zhao et al., 2023) explore resolution adaptation in generative modeling and could strengthen the discussion around DiT-DMP.

**Experimental Designs Or Analyses:**

Nothing to add here.

**Methods And Evaluation Criteria:**

The methods and evaluation criteria align well with the problem. The authors use relevant benchmark datasets—ModelNet40, spatial transcriptomics, and ImageNet—covering diverse structured data types. Metrics like Wasserstein distance and FID are appropriate for measuring distribution alignment and sample quality. While comparisons to baseline graph-based models are useful, evaluating against other adaptive GNN architectures could further strengthen the analysis.

**Other Comments Or Suggestions:**

No

**Other Strengths And Weaknesses:**

The paper presents a compelling and well-motivated approach to generative modeling of geometric graphs by introducing Noise-Conditioned Graph Networks (NCGNs). The core idea—that graph neural network architectures should adapt dynamically to the noise level—is both intuitive and well-supported by theoretical and empirical analyses. I found the motivation strong, particularly in demonstrating that static graph structures are suboptimal for generative processes where noise plays a fundamental role. The experiments are thorough, covering diverse domains such as 3D point clouds, spatiotemporal transcriptomics, and image generation, which strengthens the paper’s claims regarding the generality of the approach.

One of the key strengths of the paper is its blend of theoretical insight and practical implementation. The information-theoretic analysis linking noise levels to optimal message-passing radius is a valuable contribution, providing a principled foundation for the proposed method. Additionally, the empirical analysis supports these theoretical findings in a convincing way, particularly through attention weight visualizations and coarse-graining experiments. The modification of an existing state-of-the-art model (DiT) to incorporate NCGNs with minimal changes is another highlight, showcasing the practical viability of the approach.

However, I do have some concerns. The theoretical results, while interesting, rely on certain simplifying assumptions which I am not sure are in conflict with real-world data. Moreover, though the empirical validation mitigates this to some extent, it would be useful to discuss potential limitations or failure cases in scenarios with highly structured noise or irregular graph connectivity patterns. Additionally, while the authors compare different adaptive scheduling strategies for DMP, the choice of an exponential schedule is somewhat heuristic. A more systematic exploration, perhaps with learned schedules or adaptive mechanisms, could further strengthen the claims.

From a clarity perspective, the paper is generally well-written, but certain sections, particularly the theoretical derivations, could be made more accessible. The notation is sometimes dense, which may make it difficult for readers unfamiliar with flow-based generative models to follow. More intuitive explanations or visualizations of key mathematical insights could improve readability.

Overall, this is a strong and original contribution with clear significance for the field of geometric generative modeling. While there are some areas for refinement, particularly in the theoretical assumptions and clarity of exposition, the paper convincingly demonstrates that noise-adaptive graph structures offer a meaningful improvement over existing approaches.

I actually really like the idea. I think if you add some more ablations and explain the limitations of the work better, it is a very solid piece of work.

**Questions For Authors:**

See sections above.

**Relation To Broader Scientific Literature:**

The paper builds on and extends several key areas in generative modeling, geometric deep learning, and adaptive graph structures:

- Flow-Based Generative Models:
The work extends diffusion models (Ho et al., 2020) and flow matching (Lipman et al., 2022) by introducing a noise-adaptive graph structure, rather than using a static neural network across the generative process. This aligns with prior work in score-based generative modeling (Song & Ermon, 2019) but introduces a dynamic graph representation to improve expressivity.

- Graph-Based Generative Modeling:
Prior works on graph generative models (Corso et al., 2022; Xu et al., 2022) typically use fixed message-passing radii (e.g., k-NN graphs) during training. This paper challenges that assumption, showing that adjusting graph connectivity and resolution based on noise level leads to better representation learning.

- Multi-Scale Graph Representations:
The use of coarse-graining at high noise levels connects to ideas in hierarchical graph representations (Li et al., 2020) and graph signal processing (Oono & Suzuki, 2020). The paper strengthens the case that adaptive resolution improves generative modeling, a principle also explored in neural operators for PDEs.

**Theoretical Claims:**

As far as I can tell, the proofs are done really well and are correct.

---

> ### Author Rebuttal · Authors · 2025-03-27
>
> We thank the reviewer for their time, effort, and constructive feedback. We address the reviewer’s concerns and questions below:
>
> > Essential References Not Discussed
>
> Thank you for pointing out these references. We will ensure these works are discussed in the related works of the camera-ready version:
>
> **Adaptive GNNs (Li et al., 2018; Shirzad et al., 2022)**: While these works and our paper both adapt the graph structure, they address fundamentally different objectives—discriminative tasks or evaluation metrics rather than generative processes. Their adaptation mechanisms also differ from our noise-conditioned approach, as they either learn task-specific static structures or improve evaluation methods without dynamically modifying architectures based on noise levels.
>
> **Noise-aware graph learning (e.g. Luo et al., 2021; Zhang et al., 2020)**: While Luo et al. (2021) redraws edges at different noise levels, they use a *static* receptive field and resolution throughout the process (same fixed radius at all noise levels, as shown in Figure 3 of their paper), contrasting with our *dynamic* approach that systematically varies both factors. Zhang et al. (2020) focus on subsampling with full attention rather than noise-conditioning.
>
> **Hierarchical graph pooling (e.g. Ying et al., 2018; Defferrard et al., 2019)**: There are similarities between these approaches and our work as they all aim to have a multiscale view of the graph. However, these methods employ fixed hierarchical structures for classification/regression tasks, while our approach continuously adapts the hierarchical representation based on the noise level of the generative model. Further, the pooling techniques in these works could complement NCGNs as coarse-graining procedures.
>
> **Multi-resolution diffusion models (e.g. Zhao et al., 2023)**: We share a common insight with Zhao et al. (2023) and other works like cascading diffusion models: it is beneficial to have coarser representations at high noise regimes and fine-grain representations at low noise regimes. However, these works operate with discrete, manually-determined resolution stages requiring separate models for each resolution, while NCGNs provide a single continuous model with automatically adapted resolution *and connectivity*.
>
> > The theoretical analysis assumes isotropic Gaussian noise and continuous graphs, which may not hold in all real-world settings. More validation on structured or irregular noise would strengthen this claim.
>
> We acknowledge the limitations in our theoretical assumptions made in Section 3.1, but would like to clarify two key points: (1) **Isotropic Gaussian noise is the standard choice** in modern flow-based generative models, including diffusion models [1], flow-matching [2], and other variants [3,4]. This makes the isotropic Gaussian noise model assumption directly applicable to most real-world generative modeling settings. (2) The continuous graph assumption is indeed a simplification but is necessary to make the theoretical analysis tractable; to address the concerns, we specifically designed our empirical analysis in Section 3.2 to validate that our theoretical insights hold even when these assumptions are relaxed.
>
> [1] Denoising Diffusion Probabilistic Models. NeurIPS, 2020.
>
> [2] Flow Matching for Generative Modeling. ICLR, 2023.
>
> [3] Flow Straight and Fast: Learning to Generate and Transfer Data with Rectified Flow. ICLR, 2023.
>
> [4] Action Matching: Learning Stochastic Dynamics from Samples. ICML, 2023.
>
> > The choice of an exponential scheduling function is somewhat heuristic, and a deeper study of learned schedules could provide stronger justification.
>
> **Implementing learnable schedules is difficult** because the schedule is used for kNN graph construction and coarse-graining procedures, which involve non-differentiable operations (sorting, clustering, discrete assignments) that are not amenable to standard backpropagation. Instead of relying purely on heuristics, we conducted ablation studies in Table 2 and Section 5.1.1 comparing four scheduling functions (linear, exponential, logarithmic, ReLU), finding that all significantly outperform fixed baselines, with exponential consistently performing best.
>
> While developing differentiable approximations for these operations represents an exciting future direction, we believe it warrants dedicated study beyond our current paper's scope.
>
> > certain sections, particularly the theoretical derivations, could be made more accessible. The notation is sometimes dense, which may make it difficult for readers unfamiliar with flow-based generative models to follow.
>
> We agree that some theoretical sections and notation could be made more accessible. Are there any specific sections, derivations, or notations you think could be improved? This would help us prioritize changes for the camera-ready version.
>
> ---
> Thank you again for your helpful feedback. We welcome any additional discussions.

---

### Official Review · Reviewer_5xLj · 2025-03-14

**Overall Recommendation:** 3

**Summary:**

This work propose to change the architecture of the backbone model according to the noise level in flow match models. It shows that the reception field should be expanded and the resolution should be coarsen in high noise level. Based on this insight, this work proposes  DMP, which consistently outperforms noise-independent architectures on 3D point clouds, spatio-temporal data and images.

**Claims And Evidence:**

Authors somehow overclaims their contribution. In abstract, this works claims to change GNN architecture according to noise level, however, only resolution and reception fields are changed, while other architecture designs are not discussed.

**Essential References Not Discussed:**

In section 3.1, this work cited no work on geometric generative models.

**Experimental Designs Or Analyses:**

Yes, I checked section 5.1.

**Methods And Evaluation Criteria:**

This work includes various tasks. However, the baseline only includes the most vanilla models in the field, and the SOTA methods are not compared.
Moreover, resolution/reception field in the center point in this work. However, graph transformers and GNNs with virtual node are known to capture global information. This work includes DiT in image task, but these global GNN baselines are still important for other tasks.

**Other Comments Or Suggestions:**

Table's caption should be placed above the table.

**Other Strengths And Weaknesses:**

Clear theory and straightforward intuition on the relation between resolution and noise level.

**Questions For Authors:**

Will the change of architecture and graph during generation leads to significant computation overhead?

**Relation To Broader Scientific Literature:**

This work improves flow-based generative models. But its idea is novel to me.

**Theoretical Claims:**

Yes, I checked Section 3.

---

> ### Author Rebuttal · Authors · 2025-03-27
>
> We thank the reviewer for their time, effort, and constructive feedback. We address the reviewer’s concerns and questions below:
>
> > In section 3.1, this work cited no work on geometric generative models
>
> Our paper references key geometric generative modeling works in the introduction (e.g. [1-5]), but we agree that these references should be more explicitly discussed in Section 2.1 as well. We will ensure these important works are properly cited and discussed in Section 2.1 of the camera-ready version. Please let us know if there are relevant references we missed.
>
> [1] An autoregressive generative model of 3d meshes. ICML, 2020.
>
> [2] Pointflow: 3d point cloud generation with continuous normalizing flows. ICCV, 2019.
>
> [3] Highly accurate protein structure prediction with alphafold. Nature, 2021.
>
> [4] Diffdock: Diffusion steps, twists, and turns for molecular docking. ICLR, 2023.
>
> [5] Spatiotemporal modeling of molecular holograms. Cell, 2024.
>
> > Authors somehow overclaims their contribution. In abstract, this works claims to change GNN architecture according to noise level, however, only resolution and reception fields are changed, while other architecture designs are not discussed.
>
> Our original claim is for **the general framework of NCGN** since it is the first to formalize conditioning the GNN architecture on the noise-level of the generative process. **DMP is a specific implementation** of NCGNs that provides a practical model that can be used out-of-the-box with current flow-based generative models, as shown in Section 5.3.
>
> And while it is true that many other architectural components of GNNs could be conditioned on the noise level, Section 3 of our paper theoretically and empirically suggests that resolution and reception are two critical changes that impact expressivity, which is why we focus on these in our paper. Thus, our work serves as a first step in the potentially many implementations of NCGNs to improve performance of geometric generative models. As noted in the conclusion, NCGNs could adapt other architectural components (layer count, width, message passing type) in future work. We will clarify this distinction more explicitly in the camera-ready version to prevent any perception of overclaiming.
>
> > Baseline only includes the most vanilla models in the field, and the SOTA methods are not compared. Moreover, resolution/reception field in the center point in this work. However, graph transformers and GNNs with virtual node are known to capture global information.
>
> **DMP complements rather than replaces SOTA methods**. Our primary contribution is the noise-conditioning framework that can enhance existing architectures. In fact, DMP already incorporates concepts similar to what the reviewer suggests:
>
> 1. **DMP utilizes virtual nodes by design** - Our coarse-grained "supernodes" in Section 4.1 function similarly to virtual nodes, but with a critical difference: they dynamically adapt their number and connectivity based on noise level. At high noise, we use fewer supernodes with wider receptive fields; at low noise, we use more supernodes with localized connectivity.
>
> 2. **Our baselines effectively generalize many SOTA approaches** - Our fully-connected GAT baseline is effectively a graph transformer, the long-short range baselines capture similar connectivity patterns as architectures with static virtual nodes, and Section 5.3 demonstrates that incorporate noise-conditioning to a SOTA transformer-based model results in significant performance gains.
>
> > Will the change of architecture and graph during generation leads to significant computation overhead?
>
> Detailed in Section 4.1 under “Linear Time Complexity,” DMP maintains **linear-time message passing** throughout the generative process by ensuring that the product of connectivity ($r_t$) and resolution ($s_t$) remains constant: $r_ts_t = r_1N$. This design enables DMP to take "the best of both worlds" of having the expressivity benefits of more complex (e.g. fully connected) architectures while having the time complexity of sparsely connected methods.
>
> ---
> Thank you again for your helpful feedback. We welcome any additional discussions.

---

### Decision · Program_Chairs · 2025-05-01

**Decision:**

Accept (poster)

**Comment:**

This paper proposes Noise-Conditioned Graph Networks (NCGNs), which adapt graph architecture based on noise level during generative modeling. The main instantiation, Dynamic Message Passing (DMP), adjusts receptive field and resolution and is theoretically motivated and empirically validated across several domains.

Reviewers found the paper novel, with clear theoretical grounding and strong empirical results. Concerns included missing comparisons to some SOTA methods, overclaiming in the abstract, and the use of heuristic (non-learned) schedulers. The authors addressed these points in detail, and reviewers were generally satisfied.